



# Turbulence in the stratified boundary layer under ice: observations from Lake Baikal and a new similarity model

Georgiy Kirillin[1], Ilya Aslamov[2], Vladimir Kozlov[3], Roman Zdorovennov[4], and Nikolai Granin[2]

[1]Deaprtment of Ecohydrology, Lebiniz-Institute of Freshwater Ecology and Inland Fisheries (IGB), Berlin, Germany
[2]Department of Hydrology and Hydrophysics, Limnological Institute, Siberian Branch of Russian Academy of Sciences, Irkutsk, Russia
[3]Institute for System Dynamics and Control Theory, Siberian Branch of Russian Academy of Science, Irkutsk, Russia
[4]Northern Water Problems Institute (NWPI), Karelian Research Center, Russian Academy of Sciences, Petrozavodsk, Russia

**Correspondence:** Georgiy Kirillin (kirillin@igb-berlin.de)

**Abstract.**

Seasonal ice cover on lakes and polar seas creates seasonally developing boundary layer at the ice base with specific features: fixed temperature at the solid boundary and stable density stratification beneath. Turbulent transport in the boundary layer determines the ice growth and melting conditions at the ice-water interface, especially in large lakes and marginal seas, where large-scale water circulation can produce highly variable mixing conditions. Since the boundary mixing under ice is difficult to measure, existing models of ice cover dynamics usually neglect or parameterize it in a very simplistic form. We present first detailed observations on mixing under ice of Lake Baikal, obtained with the help of advanced acoustic methods. The dissipation rate of the turbulent kinetic energy (TKE) was derived from correlations (structure functions) of current velocities within the boundary layer. The range of the dissipation rate variability covered 2 orders of magnitude, demonstrating strongly turbulent conditions. Intensity of mixing was closely connected to the mean speeds of the under-ice currents, the latter being of geostrophic origin and having lake-wide scales. Mixing developed on the background of stable density (temperature) stratification, which affected the vertical structure of the boundary layer. To account for stratification effects, we propose a model of the turbulent energy budget based on the length scale incorporating the dissipation rate and the buoyancy frequency (Dougherty-Ozmidov scaling). The model agrees well with the observations and yields a scaling relationship for the ice-water heat flux as a function of the shear velocity squared. The ice-water heat fluxes in the field were the largest among all reported in lakes (up to 40 W m$^{-2}$) and scaled well against the proposed relationship. The ultimate result consists in a strong dependence of the water-ice heat flux on the shear velocity under ice. The result suggests large errors in the heat flux estimations, when the traditional "bulk" approach is applied to stratified boundary layers. It also implies that under-ice currents may have much stronger effect on the ice melt than estimated by traditional models.



# 1 Introduction

The demand on a better quantitative description of the formation, evolution, and decay of the seasonal ice has grown recently because of large-scale trends to shortening of the ice season in the Northern Hemisphere and the drastic decrease of the arctic sea ice extent. Closure of the global mass budget of the arctic seasonal ice is a complex problem, related, apart from the

atmospheric and terrestrial heat sources, to the upward transport of heat stored in the under-ice water body. An important role in the heat budget of seasonal ice is played by the storage of the solar radiation in the under-ice water, which is subsequently transported to the ice base by the under-ice currents. The effect of currents on ice melt is particularly apparent in the Arctic ocean, where the loss of ice mass in spring and summer occurs mainly from the ice bottom (McPhee, 1992; Perovich et al., 2011; Carmack et al., 2015; Peterson et al., 2017). Apart from the polar oceans and seas, seasonal formation of ice cover is

an essential feature of high-latitude freshwater lakes. Physics of seasonal ice cover on lakes has gained particular attention, as an essential part of climate change research (Magnuson et al., 2000; Kirillin et al., 2012). A shorter seasonal ice cover as a result of global warming may produce a positive feedback due to increase of greenhouse gas emission and changing the global carbon budget (Tranvik et al., 2009). Hence, estimation of the consequences of phenological changes on inland waters requires quantification of the physical mechanisms that control the formation and melting of ice. The heat and mass transfer at the

ice-water interface is the least studied among these mechanisms (Kirillin et al., 2012; Aslamov et al., 2014).

The seasonal ice cover on lakes, especially, on large ones, shares many basic features with the seasonal sea ice. Storage of the heat from solar radiation in the surface mixed layer (SML) and its subsequent release to the ice base is the major mechanism of the ice cover melt in lakes (Kirillin et al., 2012) as well as in the ocean (Perovich and Richter-Menge, 2009). However, in contrast to the seawater, lakes own some specific physical features affecting formation and melting of ice. Water temperatures

in ice-covered freshwater lakes are below their value of maximum density. Therefore, solar heating of upper layers produces free convection, which is the major mechanism of the SML formation (Mironov et al., 2002). In addition to the storage of the heat from the short-wave radiation penetrating the ice, convective mixing in the SML entrains the warmer water from the deep layers. The convective SML is separated from the ice base by a stably stratified interfacial layer (IL) with the upward temperature drop down to the freezing point of freshwater. The strong stratification in the IL prevents convective transport of

heat to the ice-water interface. As a result, only a small amount of the heat is available for ice melt, despite strong convection in the SML (Kirillin et al., 2018b). The situation is akin to formation of a stably stratified layer beneath the ice base and the near-surface temperature maximum in marginal polar seas, driven by freshening of the surface waters due to river runoff or the accelerated ice melt (Jackson et al., 2010, 2012).

The ice-water interaction becomes more complex when a freshwater lake becomes essentially large compared to the Rossby

radius of deformation (Gill, 1982). The latter condition suggests long-lasting water circulation under ice, which, similarly to the ocean circulation, is able to produce significant velocity shear at the ice base and accelerate by this the upward heat transport (McPhee, 1992). Among such lakes, Lake Baikal—the largest lake by volume on earth—most closely resembles the Arctic Ocean with regard to the seasonal ice dynamics. Thanks to the strong winter cooling under influence of the Siberian atmospheric pressure maximum, Lake Baikal reveals a steady ice cover over the entire lake for 3-5 months of the year. Consequently, the





seasonal ice regime plays a crucial role in hydrodynamics and ecosystem functioning of the lake. Aslamov et al. (2014) reported high heat fluxes from water to ice of Lake Baikal during the period of ice growth. The fluxes were apparently related to the water circulation pattern beneath the ice cover. Despite the thin snow cover, the convection due to penetrating solar radiation was not able to produce such a strong boundary mixing, exceeding the fluxes measured in small lakes (Kirillin et al., 2018b) up

to an order of magnitude. Hence, the high turbulence level was tentatively referred by the authors to the shear mixing produced by the water circulation under the ice surface.

The intensity of turbulence produced by velocity shear in the boundary layer and the resulting heat transport from water to the ice base may vary depending on the current velocity, ice structure, and density stratification under ice. In order to estimate the effect of under-ice circulation on the ice-water heat flux in lakes, we performed a field experiment combining temperature

measurements with high temporal and vertical resolution within the ice cover and fine-scale registration of current velocities under the ice base. The temperature observations were subsequently used for estimation of the heat budget at the ice-water interface and derivation of the ice-water heat fluxes. The data on fine-scale velocity fluctuations provided information on variability of mean currents under ice as well as on the characteristics of turbulent mixing in the ice-water boundary layer in form of the dissipation rate of the turbulent kinetic energy (TKE). Below, both outcomes of the field experiment are combined

to reveal the characteristics of the turbulent boundary layer and to analyze the effect produced by turbulent mixing on the ice cover thickness. The overarching goal of the presented study consists in establishing the scaling relationships linking the under-ice circulation and the seasonal ice cover dynamics suitable for parameterization of the ice-water heat exchange in regional and global models of seasonal ice.

## 2  Heat budget of seasonal ice cover and scaling of the under-ice boundary layer

To a good approximation, the base of the lake ice can be represented as a rigid boundary on top of a fluid, i.e. the vertical heat transport at the ice-water interface is close to purely conductive on both ice and water sides, governed by molecular forces within the ice cover and within a thin "conduction" layer of water. It should be noted that the assumption holds generally true for a solid freshwater ice with low amount of impurities: Salt-water ice undergoes brine extraction, which can induce convection by mass flux at the boundary and increase remarkably the heat transport. Similar increase of water flow and destruction of the

conduction layer at the ice-water interface may take place in a "rotten" freshwater ice subject to internal melting, especially in presence of impurities (Bluteau et al., 2017). In the majority of freshwater lakes the aforementioned effects are negligible during the most of the ice season. In particular, in Lake Baikal, due to cold winters and low snow precipitation, practically 100% of the ice cover consists of clear congelation ("black") ice, which grows at the ice-water interface, has homogeneous crystal structure and much lower amount of impurities than the sea ice or the river ice (Kirillin et al., 2012). Hence, the heat

balance at the ice-water interface (IWI) can be expressed as the sum of conductive (molecular) boundary fluxes $Q_{cw}$ and $Q_{ci}$ and the heat release/consumption due to the phase change (freezing or melting) (Aslamov et al., 2014):

$$Q_{cw} = Q_{ci} - \rho_i L_f \frac{dh_i}{dt} \qquad (1)$$





where the vertical coordinate is directed downwards with the origin $z_i$ at the IWI, $dh_i dt^{-1}$ is the rate of basal ice melting (growth); $\rho_i L_f$ is the product of the ice density and latent heat of fusion, $Q_{cw}$ is the conductive heat flux from/to the water,

$$Q_{cw} = C_{pw} \rho_w \kappa_w \left. \frac{dT}{dz} \right|_{z_i+0}, \tag{2}$$

and $Q_{ci}$ is the conductive heat flux from/to the ice,

$$Q_{ci} = C_{pi} \rho_i \kappa_i \left. \frac{dT}{dz} \right|_{z_i-0}. \tag{3}$$

Temperature at the IWI is fixed at the melting point of $0$ °C that corresponds to the following thermodynamic characteristics: molecular heat diffusion coefficient for fresh water $\kappa_w \approx 1.4 \cdot 10^{-7}$ m$^2$ s$^{-1}$, molecular heat diffusion coefficient for ice $\kappa_i \approx 1.1 \cdot 10^{-6}$ m$^2$ s$^{-1}$; the product of the water heat capacity and density is $C_{pw}\rho_w \approx 4.18 \cdot 10^6$ J K$^{-1}$ m$^{-3}$, and the same product for ice is $C_{pi}\rho_i \approx 1.96 \cdot 10^6$ J K$^{-1}$ m$^{-3}$ (see e.g. Leppäranta, 1983).

Equation (1) can be applied for reliable estimation of the ice-water heat flux $Q_{wi}$ if the temperature profile within the ice cover and the time variations of the ice thickness $dh_i dt^{-1}$ are known. This approach was used for estimation of the heat fluxes in Lake Baikal by Aslamov et al. (2014), who recorded the temperature profile within the ice cover and the variations of the ice thickness $dh_i dt^{-1}$ with high temporal resolution.

However, direct estimation of $Q_{cw}$ in the absence of detailed data on the ice cover dynamics and temperature is less straight-forward. The bulk of the water column under the ice is turbulent: While ice-covered waters are isolated from the direct influence of wind, vertical heat transport remains higher than the purely molecular one, intensified by convective mixing due to solar radiation penetrating the ice cover and due to shear turbulence produced by under-ice currents. As a result, the thickness of the "diffusive" layer in the immediate vicinity of the ice base, where Eq. (2) holds true, does typically not exceed several millimetres. The temperature gradient $dT dz^{-1}$ at $z = 0$ is barely detectable by the traditional observation methods and is varying continuously depending on the mixing and temperature conditions in the underlying water column.

The turbulent kinetic energy (TKE) under ice is supplied by the decay of the convective motions in the underlying convectively mixed layer (Mironov et al., 2002) and/or by mean horizontal circulation (McPhee, 1992; Aslamov et al., 2014). In the latter case, the vertical turbulent transport of momentum $\tau = <u'w'> = u_*^2$ is created by the current velocity shear $S = \partial U_{\text{mean}} \partial z^{-1}$ at the ice base. Hence, close to the IWI, the distance from the ice base $z$ is a major parameter, determining the turbulent mixing characteristics. Assumption of the proportionality between the mixing length scale and the distance from the solid boundary leads to the relationships for the neutral (logarithmic) boundary layer,

$$S = \frac{\partial U_{\text{mean}}}{\partial z} = \frac{u_*}{\kappa z}, \tag{4}$$

or

$$U_{\text{mean}}(z) = \frac{u_*}{\kappa} \ln\left(\frac{z}{z_0}\right), \tag{5}$$

where $z$ is the turbulent pulsation length scale equal to the distance from the lower boundary of the ice, $z_0$ is the roughness parameter of the ice bottom, $\kappa \approx 0.4$ is the empirically determined von Kármán constant, $u_*^2$ is the turbulent stress (shear velocity squared) produced by the vertical shear of the mean velocity $S$.





The logarithmic velocity distribution in the ice cover vicinity makes possible estimation of the momentum flux based on mean velocity values only using a direct relationship derived from Eq. (5) ("bulk" formula),

$$u_*^2 = C_Z U_z^2, \tag{6}$$

where the bulk transfer coefficient $C_Z$ corresponds to the depth $Z$ of the current speed measurements, and is defined as

$$C_Z = \left( \frac{\kappa}{\log z - \log z_0} \right)^2. $$

For a non-stratified steady-state turbulent boundary layer, the TKE budget tends to the local balance of the largest terms in the TKE transport equation, production and dissipation:

$$\varepsilon = u_*^2 S \propto \frac{u_*^3}{z}, \tag{7}$$

where $\varepsilon$ is the TKE dissipation rate.

The second factor influencing the buoyancy flux at the IWI—the destabilizing buoyancy flux $B_R$ due to volumetric absorption of solar radiation $I(z)$ within the convectively mixed water column of thickness $h_S$—is derived from the heat transport equation with radiation term as

$$B_R = \beta \left( I(0) + I(h_S) - 2 h_S^{-1} \int\limits_0^{h_S} I(z) dz \right). \tag{8}$$

Here, the assumption of height-constant warming rate within the convective layer was used (Mironov et al., 2002). $\beta = g \alpha_T(T)$
is the buoyancy parameter, $\alpha_T$ is the thermal expansion coefficient. The latter is generally not constant in freshwater due to non-linearity of the equation of state at temperatures close to the maximum density value $T_m d \approx 3.98$ °C [$\alpha_T(T) \approx 0.825 \cdot 10^{-5}(T - T_m d)$ K$^{-1}$, see e.g. Farmer and Carmack, 1981].

     The ice-water boundary layer in freshwater lakes is rarely neutrally stratified: A distinctive feature of the layer is the fixed temperature at the IWI. As a result, the water adjacent to the IWI in fresh or brackish environments is always subject to stable
stratification, with deeper waters being warmer and thereby heavier. Stratification may alter the turbulent length scale, affecting Eqs. (4)-(7). Stratification counteracts the shear production of turbulence, and in the asymptotic case of a strongly stratified layer, is the sole mechanism of turbulence damping. This effect can be accounted for by introduction an additional length scale apart from the distance to the boundary $z$, as expressed by a simple formula following from dimensional analysis,

$$S = \frac{u_*}{\kappa z} \left( 1 + C_x \frac{z}{L_x} \right), \tag{9}$$

where $L_x$ is the stratification length scale and $C_x$ is an empirical coefficient. Eq. (9) replaces Eq. (4) in stratified conditions with corresponding changes in Eqs. (5)-(7). For conditions dominated by the stabilizing buoyancy flux at the boundary $B_s = \beta Q_{cw}(c_{pw}\rho_w)^{-1}$, the stratification length scale turns to the well-known Monin-Obukhov length scale $L_{MO} = u_*^3 B_s^{-1}$, with empirically determined coefficient $C_x \approx 5$ (Stull, 2012), building the basis for the Monin-Obukhov similarity theory (MOST).





If stratification is created outside the boundary layer, its effect on boundary mixing is independent on the surface buoyancy flux. A characteristic length scale for turbulence in stratified media was independently proposed by Dougherty (1961) and Ozmidov (1965), as

$$L_N = \varepsilon^{1/2} N^{-3/2}, \tag{10}$$

where

$$N = \left( \frac{g}{\rho} \frac{\partial \rho}{\partial z} \right)^{1/2} \approx \left( \beta \frac{\partial T}{\partial z} \right)^{1/2}$$

is the buoyancy frequency, $\rho$ is water density in assumption of negligible salinity effects. Replacement of $L_x$ by $L_N$ in Eq. (9) yields in this case

$$S = \frac{u_*}{\kappa z} \left( 1 + C_N \frac{z N^{3/2}}{\varepsilon^{1/2}} \right), \tag{11}$$

with the corresponding expression for the TKE production rate $P$ (cf. Eq. 7)

$$P = u_*^2 S = \frac{u_*^3}{\kappa} \left( \frac{1}{z} + \frac{C_N}{L_N} \right). \tag{12}$$

Close to the boundary, $z << L_N$, Eq. (11) approaches the neutral scaling relationship (4). At large distances from the boundary, $z >> L_N$, Eq. (11) turns to a "z-less" scaling

$$S = C_N \frac{u_*}{\kappa L_N}, \tag{13}$$

which in turn yields the $N$-scaling for $P$

$$P \propto u_*^3 L_N^{-1}.$$

In stably stratified conditions, production of TKE is balanced by two major processes, dissipation $\varepsilon$ and work against the stability forces $B_{St}$. The latter can be expressed in form $B_{St} = K_\rho N^2$, where $K_\rho$ is the diapycnal diffusivity. From analysis of dimensions, the turbulent diffusivity can be scaled as $K_\rho \propto u_*^2 N^{-1}$ (see e.g. Monin and Ozmidov, 1981). Then, the TKE

budget can be approximated as

$$\frac{u_*^3}{\kappa} \left( \frac{1}{z} + \frac{C_N}{L_N} \right) = C_B u_*^2 N + \epsilon, \tag{14}$$

with coefficients $C_N$, $C_B$ subject to estimation from empirical data, or

$$u_*^2 N \left( Ri^{-1/2} - C_B \right) = \varepsilon, \tag{15}$$

where $Ri$ is the gradient Richardson number,

$$Ri = \frac{N^2}{S^2}, \tag{16}$$





expressing the relative importance of stratification and velocity shear for the vertical transport. Its critical value $Ri_{cr} \approx 1/4$ (Turner, 1979) marks the boundary between turbulent conditions, at which the shear can destroy the stratification and the quiet conditions, at which strong $N$ ultimately suppresses any turbulent motions. Hence, for turbulence to exist at weakly supercritical $Ri$, it is required $0 < C_B < 2$ in Eq. (15). In the following we tentatively assume $C_B \approx 1$. Another scaling relationship

relevant to the turbulent mixing on the background of stable stratification is the buoyancy Reynolds number,

$$Re_b = \frac{\varepsilon}{\nu N^2}, \tag{17}$$

where $\nu = \mathcal{O}(10^{-6})$ m$^2$s$^{-1}$ is the water viscosity. $Re_b$ refers to the work of turbulence against stratification and viscosity, which becomes important at distances from the solid boundary shorter than characteristic length scales of turbulence; its critical value is reported to be $\mathcal{O}(10^1)$ (Gargett et al., 1984).

In neutral conditions, the coefficient of turbulent heat transfer $K_Z \propto u_* z$ (assuming the turbulent Prandtl number is approximately 1), and the corresponding relationship for the ice-water heat flux $Q_{iw}$ (Eq. 1) can be written as:

$$Q_{iw} \propto u_* \Delta T \propto \frac{u_* h_T}{g \alpha_T} N^2, \tag{18}$$

where $\Delta T$ is the temperature difference across the layer $h_T$ beneath the ice base, often assumed in models as water temperature at the vertical grid point closest to the ice. The expression is sometimes used in form of the "bulk" formulation, assuming direct

relationship between the friction velocity and the main current speed $u_* \propto U$ (cf. Eq. 6):

$$Q_{cw} = C_Q U \Delta T, \tag{19}$$

where $\Delta T$ is the temperature difference across $h_T$, and $C_Q$ is an empirical "bulk" heat transfer coefficient. The values of $C_Q$ were reported in the range $[0.8 \pm 0.3] \cdot 10^{-3}$ (Hamblin and Carmack, 1990; Nan et al., 2016); stratification effects on $C_Q$ were not investigated.

Adopting the same scaling considerations as in Eqs. (14)-(15), the heat flux at the IWI $Q_{iw}$ in strongly stratified conditions may be assumed to depend on the work of turbulence against the stability,

$$Q_{iw} = K_\rho \frac{\partial T}{\partial z} \propto u_*^2 N^{-1} \Delta T h_T^{-1},$$

or, in terms of buoyancy flux $B_{iw}$,

$$B_{iw} = g \alpha_T Q_{iw} \propto u_*^2 N. \tag{20}$$

Herewith, a strongly stratified case is characterized by the flux dependence on the shear velocity squared and a weaker dependence on the stratification, expressed by $1/2$ exponent at the vertical density gradient (as revealed by the direct proportionality to the buoyancy frequency $N$).

Summarizing the considerations above, validation of the Dougherty-Ozmidov scaling (Eqs. 10-14) for the ice boundary layer, and the ice-water flux parameterization (Eqs. 18-20) is possible when field data are available on both the TKE dissipation rates

and the mean fields of governing forces (production of convective instability by radiative heating and/or mean horizontal flow due to under-ice currents).




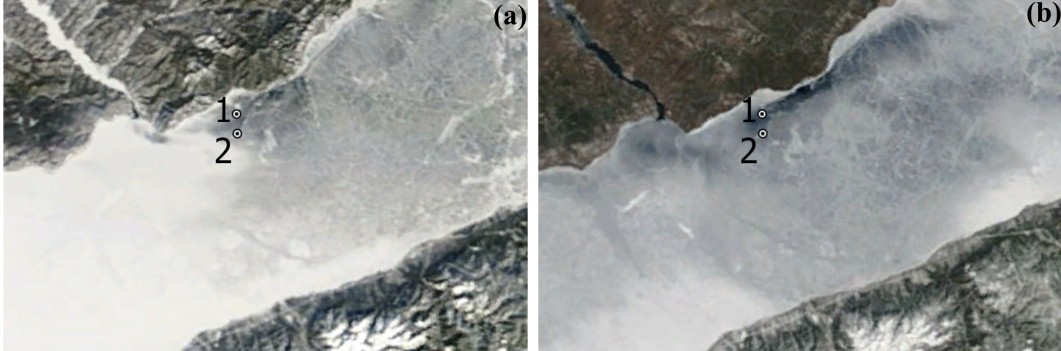

**Figure 1.** Ice conditions in Southern Baikal on 9 February (a) and 12 April 2017 (b) and locations of the autonomous measurement stations. The satellite images are from the Irkutsk Center of Remote Sensing (sputnik.irk.ru, 2019). Note the stronger ice melt in the area of the jet current around Station S1, visible as a dark area in Panel b.

## 3 Study site and field methods

The field study was performed in February-March 2017 in the southern part of Lake Baikal. Two custom-made autonomous stations were installed in the vicinity of a quasi-stationary longshore current, which has been regularly observed in this region during ice cover period (Fig. 1).

Station S1 was installed 4.5 km away from the lake shore (51° 51.923'N, 105° 4.779'E) in the area of quasi-stationary jet-like alongshore current. Station S2 was located 3.5 km to the south from station S1. The total water depth in the vicinity of both stations amounted at $\approx 1600$ m. Each station registered temperature at 30 vertical levels distributed within the ice cover, the water boundary layer, and the air above the ice. The distance between temperature sensors was 5 cm within the ice and in the under-ice boundary layer, increasing up to 10-50 cm at larger distances from the ice boundary in the water and in

the air. Three sensors of the short-wave solar radiation registered the vertical radiation decay within the air-ice-water system. Ice thickness was measured by a 330-kHz echo-sounder, deployed upward-looking at a fixed distance from the ice surface. The resolution of the system was 0.002 °C for temperature, 0.1 W m$^{-2}$ for solar radiation, and 0.1 mm for ice thickness (the operation range of 0.2–2.8 m). The system collected data with a period of 2 min, logging them internally, and sending data several times a day via cellular network to a remote Internet-server (see Aslamov et al., 2017, for detailed description of the

ice station configuration). Two-dimensional electromagnetic current meters "INFINITY-EM" (JFE Advantech Co., Ltd.) were used to measure the current velocities: (velocity range $\pm$ 5 m s$^{-1}$, resolution 0.02 cm s$^{-1}$, accuracy $\pm$ 1 cm s$^{-1}$). The current meters were positioned at a distance of 1 m from the surface of the ice cover. Three additional current meters were deployed at Station S1 at distances 0.6, 0.8 and 1.4 m from the ice surface.

Characteristics of turbulent mixing in the under-ice boundary layer were obtained with the help of the high-resolution

Doppler current velocity profiler HR Aquadopp (Nortek AS, Norway). The profiler was deployed for 48 hours successively at each of the two stations, on 05-07 March 2017 on Station S1, on 08-10 March on Station S2. The profiler was frozen into the ice downlooking, with the acoustic head at 2 cm beneath the ice base (verified with an ROV camera). Three components of





current velocity were registered with a time interval of 2 s and a spatial resolution of 15 mm in the pulse-to-pulse coherence (high-resolution) mode.

The values of short-period fluctuations of the flow velocity were used to calculate dissipation rates of the kinetic energy of turbulence (TKE) based on the Kolmogorov's 1941 hypothesis on the self-similarity of the velocity structure functions using the method described by Wiles et al. (2006). $\varepsilon$ was derived as a coefficient in the semi-empirical equation for the velocity structure function $D_i(r)$ along the $i$-th acoustic beam,

$$D_i(r) = \text{Noise} + C_v \varepsilon^{2/3} r^{2/3}, \tag{21}$$

which includes noise estimation $\text{Noise}$ due to instrumental noise and non-turbulent velocity fluctuations. Here, the constant $C_v = 3^{1/3}$ (see e.g. Lien and D'Asaro, 2002). The velocity structure function was calculated from the measured along-beam velocities $v_i(z)$ at the distance $z$ from the instrument's head as

$$D_i(r) = \left\langle (v_i(z) - v_i(z+r))^2 \right\rangle \tag{22}$$

Quality check was performed based on values of $\text{Noise}$ in Eq. (21); the $\varepsilon$ values from three beams were compared for similarity and averaged. The detailed procedure of data post-processing and quality check is described in (Kirillin et al., 2018a; Volkov et al., 2018).

## 4  Results

### 4.1  Atmospheric conditions and ice cover thickness

The ice cover formed on Lake Baikal during the second half of January 2017 with several periods of ice break-up and re-freeze. The autonomous stations were installed on 1 Feb 2017 and provided background information on the major forces driving the ice cover development. The temperatures of the ice surface remained below the freezing point of water during the entire observations period, varying between $-14°$ C and $-2°$ C with a slight warming trend (Fig. 2a). The initial ice thicknesses were nearly the same at both stations: 23 cm at Station S1 and 24 cm (a day later) at Station S2. During the first 2 weeks of February the ice grew at a nearly constant rate of 1.2-1.3 cm day$^{-1}$ (Fig. 2b). During this period, the ice surface temperatures at both stations were nearly equal and followed closely the air temperatures at the 1.5 m height. This quasi-neutral stratification in the air-ice boundary layer lasted until the end of February, caused apparently by convective heat flux from the ice surface due to release of the latent heat of ice formation. Later, the ice thickness at Station S1 (the one with strong under-ice currents) remained nearly constant, while continued to grow at a slow rate of $\approx 0.3$ cm day$^{-1}$ at Station S2 (Fig. 2c). In mid-March, a stable stratification developed in the air above the ice with air temperatures dropping down to $-16°$ C. Whereas the temperature at the ice surface of Station S2 also decreased following the air temperature trend, the ice surface at Station S1 remained relatively warm, suggesting, together with the nearly constant ice thickness, a balance between the heat release to the atmosphere and the heat supply from the water column.

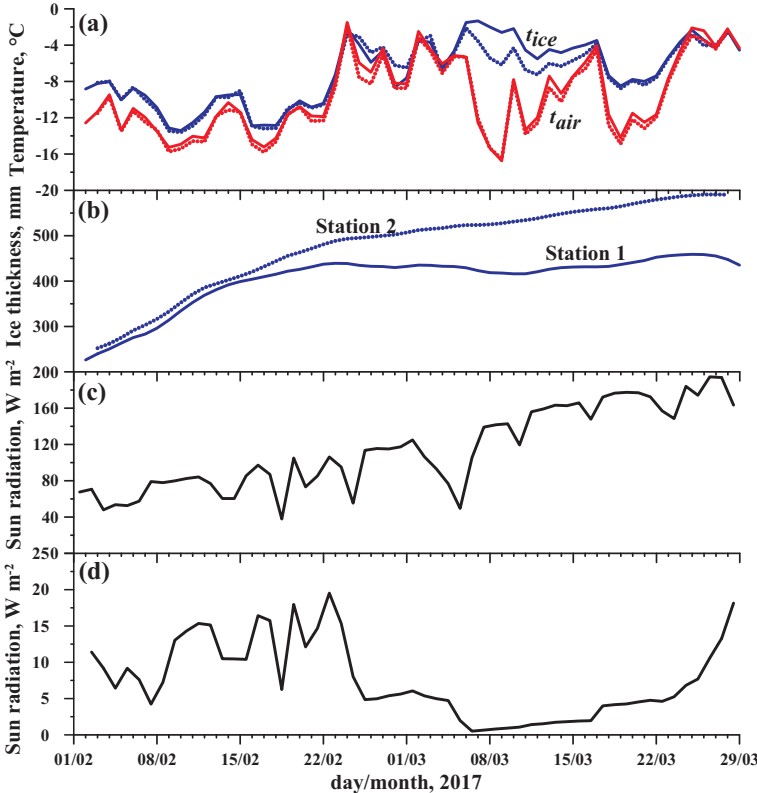

**Figure 2.** Background data on the Lake Baikal ice regime during the observations: (a) Daily averaged temperatures of ice surface ($t_{ice}$) and air temperatures at 1.5 m above the ice ($t_{air}$), (b) ice thickness, (c) incoming and (d) penetrated solar radiation. In Panels (a) and (b) solid lines correspond to Station S1 and dotted lines are for Station S2.

## 4.2 Mean currents, temperatures and stratification

Under ice, the water temperatures slightly increased with depth (Fig. 3): the mean vertical gradient of 0.6° C over the upper 10 m of the water column was about an order of magnitude weaker than those typically observed in shallow ice-covered lakes (Kirillin et al., 2018a). Below 10 m depth the water column was well-mixed vertically. Closer to the ice base, two horizontal layers could be distinguished: a $< 0.5$ m thin layer adjacent to the ice with the temperature difference of $\approx 0.3°$ C across it. Underneath, a layer with nearly linear temperature increase of $\approx 0.03°$ C m$^{-1}$ spread down to the 10 m depth.

In terms of stability, the two-layered thermal structure can be described by two nearly constant buoyancy frequencies $N_\delta \approx 2 \cdot 10^{-2}$ s$^{-1}$ in the layer $0 \leq z < \delta$ and $N_S \approx 4 \cdot 10^{-3}$ s$^{-1}$ in the layer $\delta \leq z < z_S$, where the thickness of the sub-ice layer $\delta \approx 0.4$ m and the lower boundary of the stratified interfacial layer $z_S \approx 10$ m..

The mixed layer temperature was slightly higher at S2; S1 had in turn a stronger vertical gradient close to the ice base. These temperature differences between the two stations suggested a stronger upward heat transport at S1 due to stronger vertical mixing caused by water flow. Current speeds were indeed almost twice as high at S1 than at S2 (Figs. 3, 5). The currents in



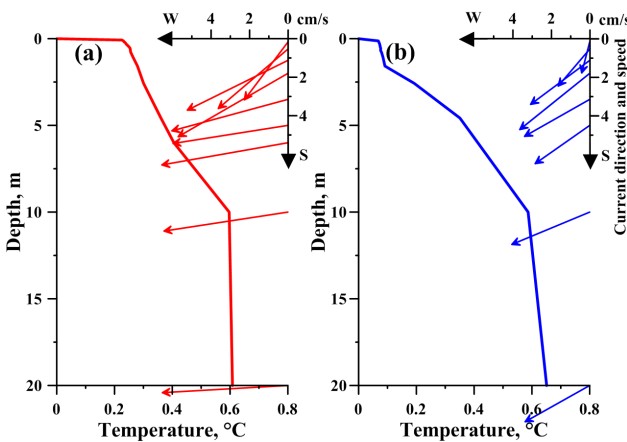

**Figure 3.** Temperature and currents velocity vector profiles: (a) Station S1, (b) Station S2.

the upper 20 m of the water column had uniform WSW direction aligned with the shoreline (see velocity vectors in Fig. 3). A weak, 10-15° anticlockwise rotation of the current vector was detectable within the 1-2 m thin layer adjacent to the ice base, suggesting some effect of the Coriolis force on the boundary layer currents. The diurnal and synoptic variations of the ice and water temperatures were similar to those observed in the previous years (Aslamov et al., 2014, 2017). The diurnal temperature

oscillations driven by the solar radiation cycle were apparent in both water column and ice cover, with amplitudes decaying towards the ice-water interface. The begin of the melt phase after 26 Mar 2017 was indicated by homogenizing of the ice cover temperature at the melting point of 0° C. Earlier, occasional increases of the air temperature, e.g. on 25 Feb, provoked deceleration of the ice growth or short-term melting periods on both stations (Fig. 4). Relevant to the matter of the present study, a remarkable increase of the ice temperatures was observed on both stations during the period of turbulence measurements on

06-12 Mar. The warming was not correlated with the air temperature: the latter dropped significantly during the same time (Fig. 2). At S1, the warming was strong enough to produce decrease of the ice thickness (Fig. 4(a)), while the effect at S2 was too weak to cause any ice melt (Fig. 4(b)).

The mean currents obtained with the acoustic Doppler profiler at a time interval of 2 s and a spatial resolution of 15 mm (Fig. 5a,b) agreed remarkably well with the records from the 5 electromagnetic velocity loggers at coarser temporal and spatial

resolution (Fig. 5a,b). The result allowed later extension of the boundary layer turbulence analysis on the whole period of electromagnetic velocity measurements, after a relationship between the mean flow characteristics and the turbulent energy production was established from the short-term acoustic profiling.

The mean current velocities from the two neighboring stations demonstrated different water flow patterns. At Station S1, current velocities in the upper 1 m of the water column had mean values $\geq 5 \cdot 10^{-2}$ m s$^{-1}$. The magnitudes underwent variations

on synoptic time scales, changing at 1 m under the ice from $\approx 10 \cdot 10^{-2}$ m s$^{-1}$ to $\approx 3 \cdot 10^{-2}$ m s$^{-1}$ within 48 hours (Fig. 5a). The event coincided with melting of the ice cover (Fig. 4(a)) suggesting the upward heat transport by the currents to be the mechanism of ice heating in this case despite the low air temperatures (Fig. 2a). Farther from the lake shore, at Station S2,





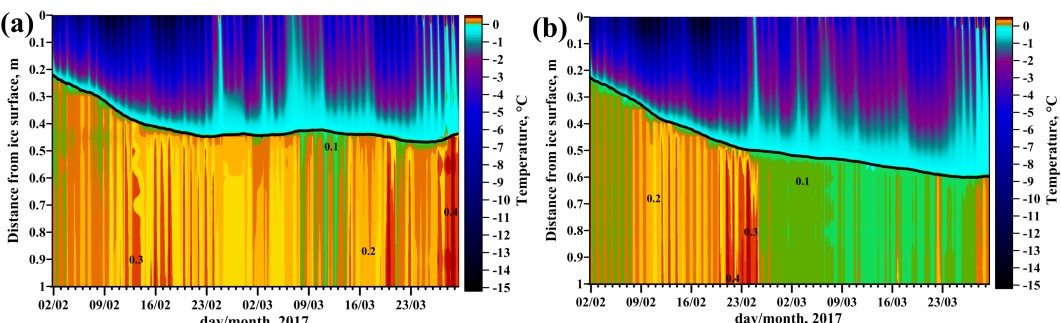

**Figure 4.** Temperature maps in the ice boundary layer and within the ice cover at Station S1 (a) and Station S2 (b). Note the different color scales for ice and water.

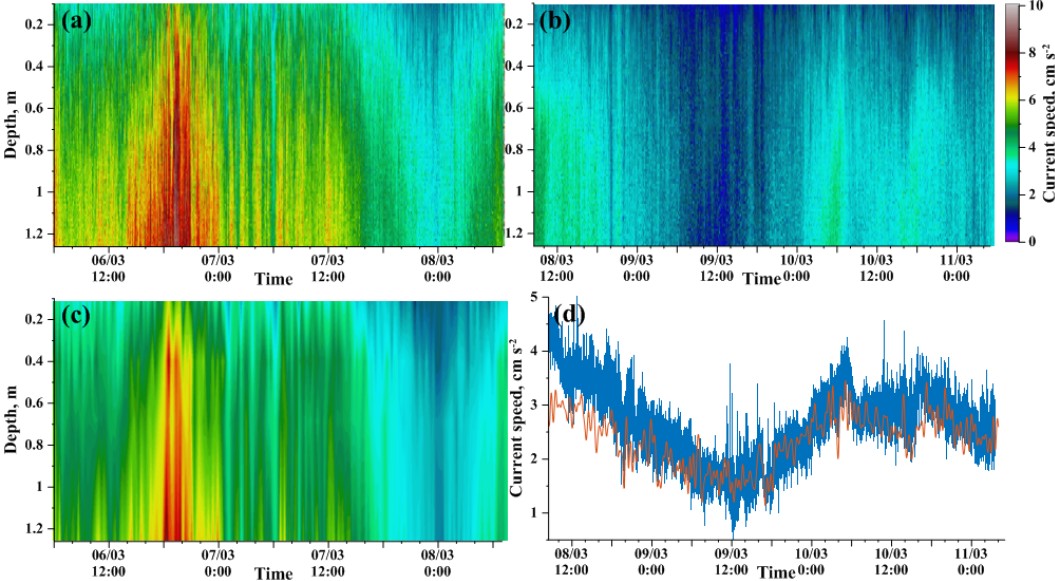

**Figure 5.** Horizontal current speeds at Station S1 (a, c) and Station S2 (b, d) measured by the acoustic Doppler profiler Aquadopp (a, b) and the electromagnetic loggers INFINITY (c). Panels (a-c) are the time-depth maps, Panel (d) shows the horizontal flow speed measured by a single INFINITY logger at 1 m under ice (thin red line) and the Aquadopp velocity record from the same depth (thick blue line).

the currents, as measured during the next 2 days, revealed a lower variability with time, and had lower magnitudes of 1 to $4 \cdot 10^{-2}$ m s$^{-1}$.

## 4.3 Solar radiation

The solar radiation flux at the ice surface doubled within the 2 months of observations (2c) contributing to the deceleration
5 of the ice growth. The light conditions under ice were estimated from continuous measurements of photosynthetically active





radiation (PAR) $I(z)$ at the single depth $z = 1.5$ m under the ice surface assuming one-band exponential decay of radiation flux (Beer Law) $I(z) = I_0 exp(-\gamma z)$. We estimated the decay rate of radiation within the water column (light attenuation coefficient) $\gamma$ using PAR profiles collected in previous studies in 2011, using the evidence that year-to-year variations of water transparency of Lake Baikal are small (Hampton et al., 2008). The light attenuation coefficient was estimated from 9 PAR

profiles as $\gamma \approx 0.17 \pm 0.01$ m$^{-1}$. The radiation flux at the ice bottom amounted at $\approx 8-18\%$ of the surface radiation, and varied depending on the snow conditions at the ice surface (2d). The mean daily under-ice short-wave radiation was $I_0 = 9.7$ W m$^{-2}$ with maximum values of up to 23.5 W m$^{-2}$.

With $\gamma$ and $I_0$ known, we estimated the theoretical thickness of the stratified interfacial layer $\delta_R$. For a single-band exponential decay of the short-wave solar radiation within the water column $I(z) = I(0) \exp(-\gamma z)$, the steady-state solution of

radiation-conduction balance in a layer of thickness $\delta_R$ can be written as (Barnes and Hobbie, 1960)

$$\gamma \kappa T_m + I_0([1 + \gamma \delta_R] e^{-\gamma \delta_R} - 1) = 0, \tag{23}$$

which represents a transcendental equation with respect to $\delta_R$. When substituted in Eq (23), the values of $I_0$ and $\gamma$ yield $\delta_R \approx 0.2$-0.4 m, adopting the temperature of the well-mixed layer of 0.6 °C for $T_m$. The estimate coincides well with the observed thickness of the ice-adjacent gradient layer $\delta$ (Fig.3). The non-zero vertical temperature gradient beneath this layer is

contrasting to the typical picture of convection in ice-covered lakes and suggests that the part of the convectively mixed layer $\delta < z < h_S$ is altered by the turbulent shear due to under-ice currents. Based on this suggestion, the under-ice radiation values were used to estimate the destabilizing buoyancy flux from Eq. (8) across the linearly stratified layer $\delta < z < h_S$ as $B_R = g\alpha I_R = g\alpha \left( I(\delta) + I(h_S) - 2h_S^{-1} \int_\delta^{h_S} I(z)dz \right)$. The resulting estimations are $I_R \approx 2$ W m$^{-2}$, and $B_R \approx 2.5 \cdot 10^{-10}$ m$^2$ s$^{-3}$. The characteristic scale of convective velocities (Deardorff, 1970) $w_* = (B_R h_S)^{1/3} \approx 1.3$ mm s$^{-1}$, which value agrees well

with previous reports on radiative convection under lake ice (Mironov et al., 2002; Kirillin et al., 2018b; Volkov et al., 2018).

### 4.4 Turbulence intensities in the ice-water boundary layer

Fluctuations of current velocities around their means were characteristic of developed turbulence: The structure functions (22) scaled well as the distance in 2/3 degree, clearly demonstrating existence of the inertial interval in the wavenumber domain (Fig. 6a). According to the 2/3-scaling, the upper boundary of the inertial interval reached $0.1 - 0.3$ m, which can be treated

as a characteristic size of turbulent eddies. In low turbulent conditions $\varepsilon < 10^{-9}$ m$^2$ s$^{-3}$, the TKE dissipation rates were at their minimum at the depth of $\approx 0.8$ m and increased towards the ice base (asterisks in Fig. 6b) supporting the scaling $\varepsilon \propto z^{-1}$ (Eq. 7). The scatter of $\varepsilon$ around the straight line $\varepsilon^{-1} \propto z$ increased with the distance from the ice $z$, starting from the $z \approx L_e$. During periods of high turbulence ($\varepsilon > 10^{-8}$ m$^2$ s$^{-3}$), the reciprocal of the TKE dissipation rate $\varepsilon^{-1}$ increased with depth more homogeneously. Nevertheless, a small local extreme in the line $\varepsilon^{-1}(z)$ and a slight change of the slope were recognizable at

the same critical distance $z \approx L_e \approx 0.8$ m from the ice (circles in Fig. 6b).

In the area with weak under-ice currents at S2, the TKE dissipation rates $\varepsilon$ varied around a value of $10^{-9}$ m$^2$ s$^{-3}$, close to the a threshold between turbulent and laminar conditions. In turn, the average $\varepsilon$ in the vicinity of the jet-like under-ice current at S1 was two orders of magnitude higher (Fig. 7). In contrast to the under-ice water temperatures, neither TKE dissipation rates

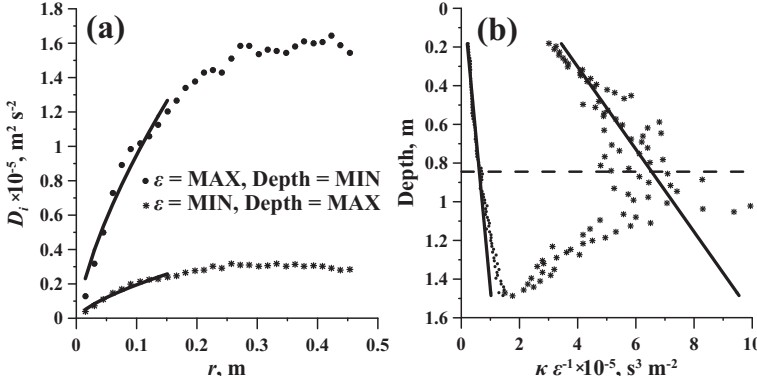

**Figure 6.** Turbulence-related characteristics of the boundary layer: (a) Velocity structure functions for high (circles) and low (asterisks) levels of TKE dissipation rates. Solid lines are the approximations by Eq. (22). (b) Vertical profiles of the reciprocal TKE dissipation rates $\kappa\varepsilon^{-1}$ for high (circles) and low (asterisks) turbulent conditions. Solid lines are the data approximations by Eq. (7). Horizontal dashed line in Panel b marks the depth equal to the mean Ozmidov length $L_N \approx 0.8$ m

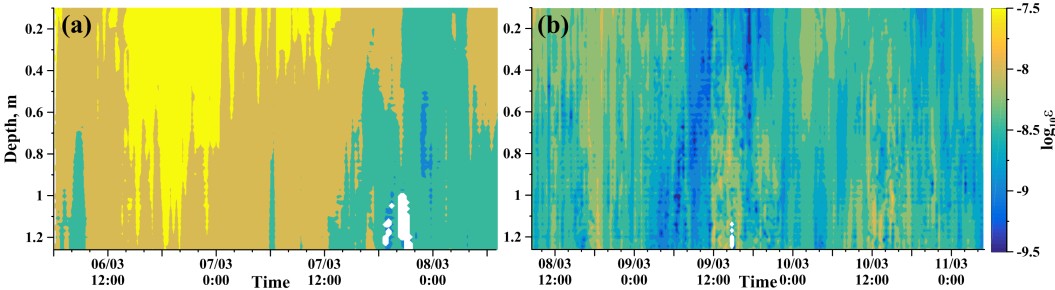

**Figure 7.** The TKE dissipation rates in the area of the jet stream (station S1, Panel a) and in the region of weak currents (station S2, Panel b).

nor friction velocities demonstrated any diurnal variations, suggesting minor effect of the radiation-driven convection on the turbulence generation. Instead, an apparent correlation existed between the turbulence intensity $\varepsilon$ and the temporal variations of the mean flow velocities (Fig. 7): the highest TKE dissipation rates of $\mathcal{O}(10^{-7})$ m$^2$ s$^{-3}$ were observed during the currents intensification up to $\mathcal{O}(10^{-1})$ m s$^{-1}$ at S1.

5    The maximum values of the Dougherty-Ozmidov length scale (Eq. 10), averaged over the period of observations, decreased with the distance from the boundary from $L_N \approx 1.5$ m at $z \approx 0.2$ m to $L_N \approx 0.9$ m at $z \approx 0.9$ m. The decrease in $L_N$ followed the decrease of $\varepsilon$. Here, the mean $N_S$ in the layer with quasi-linear stratification of 0.5-10 m beneath the ice base was used as a characteristic value of the buoyancy frequency in the Dougherty-Ozmidov (D-O) scaling. At larger distances from the ice base, $L_N$ remained nearly constant, varying between 0.8-0.9 m. Hence, the value $z_{crit} = L_N = 0.85$ can be treated as a boundary

10    between the "quasi-neutral" and strongly stratified "z-less' layers, with the turbulent length scale defined by the distance $z$ closer to the ice base, and by $L_N$ at farther distances.





An important insight into the mechanisms of turbulence generation under ice is provided by comparison of the stratification-based turbulence scaling (Eqs. 10-13) against the quasi-neutral Law-of-the-wall (LOW) relationship (Eqs. 7-5). At $z < z_{crit}$, both "neutral" relationships (4) and (7) produced similar estimations of the friction velocities $u_*$ with 20-30% higher values produced by $u_*$ estimations from $\varepsilon$ (Eq. 7). While the value of the von Kármán constant in the neutral LOW scaling $\kappa \approx 0.4$ is relatively well known from tunnel experiments and numerical simulations, and is supported by field data, the proportionality constant in Eq. (13) is not well established. Therefore, for "z-less" D-O scaling (13), the values of $u_*$ were calculated from Eq. (13) assuming a *unity* coefficient of proportionality $C_N$. In average, the "stratified" scaling produced generally lower values of $u_*$ at farther distances from the ice bottom and vice versa. The D-O scaling with $C_N = 1$ and LOW demonstrated nearly perfect agreement at $z = z_{crit} = L_N$ (Fig. 8a). The fact justified the balance between the shear producing at the boundary and the stratified production of turbulence at this distance from the wall, as well as supported the choice of the unity constant in the D-O scaling. Accordingly, $C_N = 1$ was adopted for later application of the combined log-linear scaling (Eqs. 12 and 14). The combined log-linear scaling (Eq. 12) with $C_N = 1$ produced $u_*$ close to the neutral value in the vicinity of $z_{crit}$ and decreasing towards both IWI and the open water column (Fig. 8b). Like the TKE dissipation rates, the mean current speeds demonstrated behavior characteristic of the stationary boundary layer, i.e. fitted well to logarithmic profiles at distances from the IWI less than $z_{crit}$ (Fig. 8c). Farther from the ice base the mean velocity profiles were nearly linear, with the slope close to $zL_N^{-1}$.

The integral balance between the TKE loss terms $u_*^2 N + \varepsilon$ and the turbulent energy production $u_*^2 S$ (Eq. 7) held true within the 1.5 m thick layer covered by measurements of $\varepsilon$: The mean difference between the two terms integrated over the entire layer did not exceed 0.2%; temporal variations of the dissipation rate followed closely those of the shear velocity (Fig. 9b). The balance was disturbed only at current speeds $< 0.02$ m s$^{-1}$, with a corresponding drop of $\varepsilon$ down to $< 10^{-9}$ m$^2$ s$^{-3}$ at station S2, March 9, 2017 (not shown). During this same period, the vertical flow profiles showed a significant deviation from the logarithmic form, indicating laminarization of the boundary layer under these conditions. The boundary value of the friction velocity for the transition to turbulent regime was $u_* \approx 1.0$ mm s$^{-1}$. The mean balance between the production loss terms in Eq. (14) varied however with the distance from the ice base (Fig. 9a): close to the ice-water interface the production significantly exceeded dissipation, while below the depth of $\approx 0.8$ m the dissipation prevailed above the production. Remarkably, this transition depth agreed well with the thickness of the layer where $\varepsilon \propto z^{-1}$.

The good agreement of the measured velocity profiles with the logarithmic approximation at $z < z_{crit}$ allowed estimation of the roughness of the ice bottom surface $z_0$ from Eq. (5). The mean $z_0$ amounted at 1.00 mm with maximum of 3.5 mm and minimum of 0.2 mm. The roughness had a significant (the Pearson coefficient of $-0.52$, $p << 0.01$) negative correlation with the mean velocity as $z_0 \approx 1.2 \cdot 10^{-4} U_{mean}^{-1}$.

Our estimations of $z_0$ and $u_*$ yielded the following parameters for the bulk formula Eq. (6) : $C_{1m} \approx 3.4 \cdot 10^{-3}$ and $C_Z = C_{1m} Z^{-1}$, where $C_{1m}$ is the bulk transfer coefficient for the momentum flux at 1 m depth. The independent measurements of current velocities at 4 depths made by single-point 2-D horizontal current loggers INFINITY demonstrated good agreement with Eq. (6) when scaled against $u_*$ from high-resolution AQUADOPP measurements (Fig. 10). The simple result has a large potential for application in modeling of the ice-water boundary layer at strong under-ice currents with a minimum of input



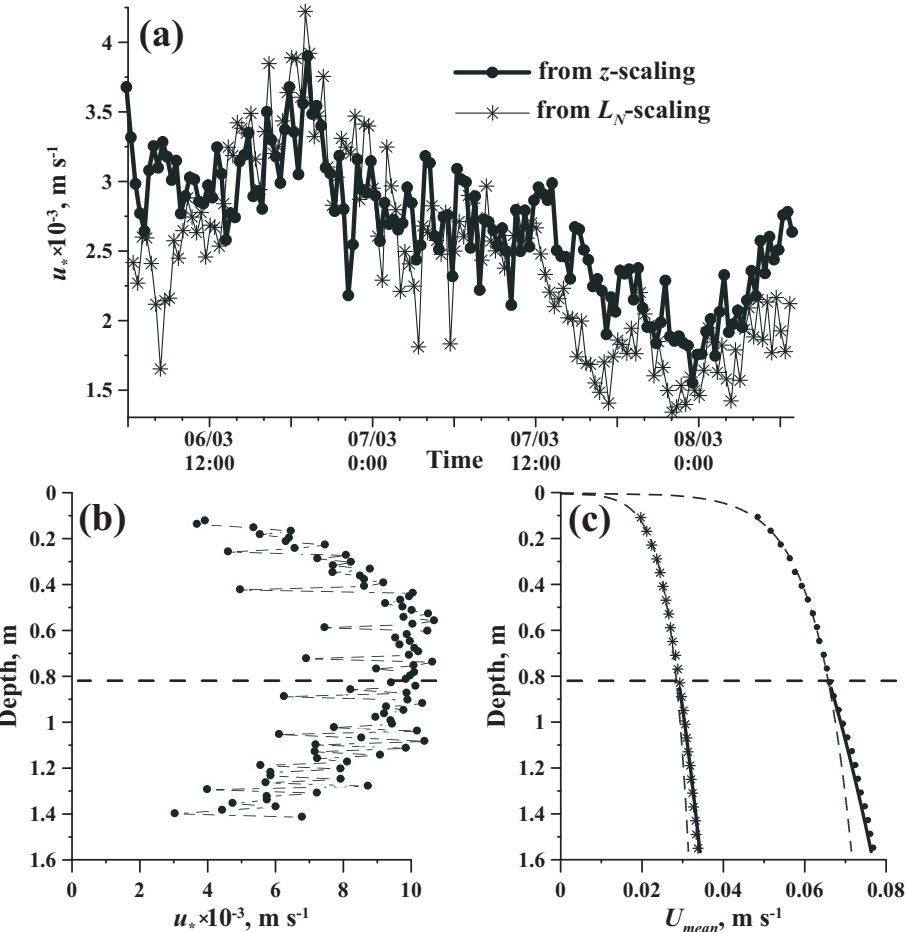

**Figure 8.** Ice boundary layer structure: (a) friction velocity $u_*$ at station S1 as determined from the quasi-neutral production-dissipation balance (Eq. 7, thick solid line with symbols) and from the Dougherty-Ozmidov length scale (Eq. 13, thin lines) at the distance from the ice base nearly equal to the mean Ozmidov length $L_N \approx 0.8$ m; (b) mean vertical profiles of $u_*$, and (c) vertical velocity profiles for strong (circles) and weak (asterisks) currents. The two profiles in Panel (c) correspond to those in Fig. 6. Horizontal dashed line in Panels (b,c) marks the depth equal to the mean Ozmidov length $L_N \approx 0.8$ m

information; a care should be taken however about the thickness of the nearly logarithmic layer and its dependence on the under-ice stratification—the effect described above and discussed in more details below.

## 4.5 Heat budget at the ice base and relation of ice-water heat flux to under-ice mixing

The heat balance at the IWI was calculated by Eqs. (1) and (3) using data on temperature within the ice cover and ice thickness variability measured by the echosounder. The ice-water heat flux was generally stronger at Station S1, correlated with stronger currents and mixing intensities. Already at the beginning of the observations period in February, the upward conduc-





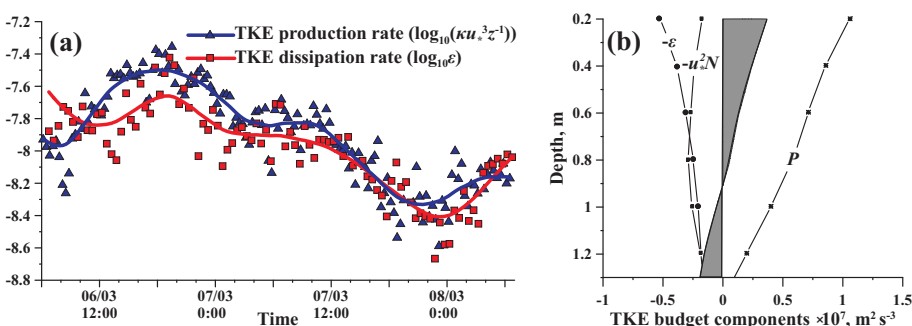

**Figure 9.** TKE production-dissipation balance in the ice boundary layer. (a) TKE production and dissipation rates at the depth of 0.85 m beneath the ice base. Bold lines are the values filtered by a moving average with a 6-hour window; (b) vertical profiles of the TKE budget components in Eq. (14) averaged over the 2 day period of measurements. Data are presented for the station S1 only; the S2 results are qualitatively the same.

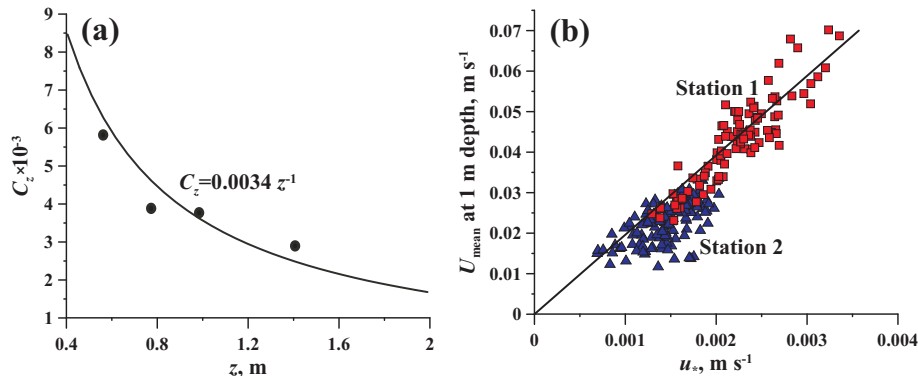

**Figure 10.** Bulk characteristics of the ice boundary layer: (a) bulk transfer coefficient for momentum $C_D$ as a function of distance from the IWI. Dots are values calculated using the horizontal velocity data from point measurements by 2-d INFINITY loggers, line is an analytical approximation; (b) relationship between the mean flow velocity at z = 1 m and the turbulent stress.

tive heat loss $Q_{ci}$ of up to 80 W m $-2$ was compensated to 30-50% by the heat supply from the water column $Q_{cw}$. The latter significantly reduced the ice growth rate and latent heat release (cf. red and blue areas in Fig. 11). During periods of currents intensification at S1 (24 Feb - 07 Mar, Fig. 11a) the heat flux from water to the ice exceeded that from the ice to the atmosphere, producing melting at the ice base (negative $Q_L$) despite surface cooling (positive $Q_{ci}$ of up to 40 W m$^{-2}$). After 25 Mar, ice cover started to melt at both stations, coinciding with an increase of air temperatures above the freezing point (Fig. 2). Quantitatively, the ice-water heat fluxes at S2 were in the range of 5-10 W m$^{-2}$, which agrees with estimations from earlier lake studies. However, $Q_{cw}$ at S1 had appreciably higher values, reaching up to 40 W m$^{-2}$ at their peaks.

——————————————————- An attempt to link the ice-water heat flux with the mixing characteristics in the stratified boundary layer in form of a bulk relationship (Eq. 19) provided a remarkable result: the heat flux at the IWI and the dissipation rate



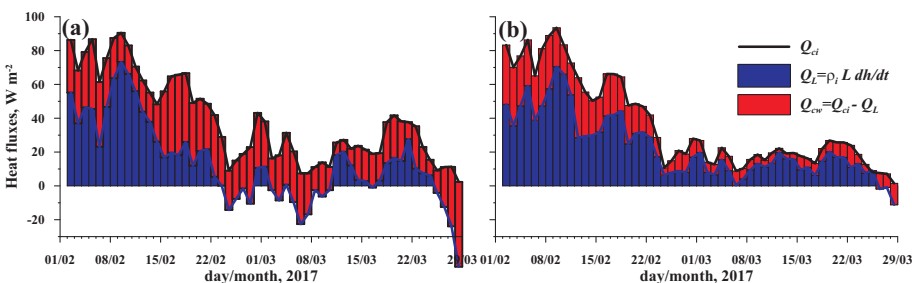

**Figure 11.** Daily averaged heat balance at (a) station S1 and (b) station S2.

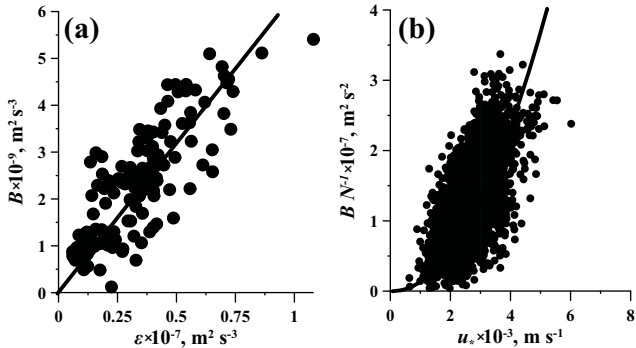

**Figure 12.** Buoyancy flux at the ice-water interface $B$ as a function of (a) TKE dissipation rate $\varepsilon$ and (b) friction velocity $u_*$

of the TKE are linked linearly (Fig. 12a) or, in terms of buoyancy flux $B$:

$$B = g\alpha Q_{cw} = 0.065\,\epsilon. \tag{24}$$

Here, $\varepsilon$ is taken at the distance $z_{crit} = 0.85$ m from the ice base that corresponds to the boundary between the ice-adjacent sublayer and the linearly stratified boundary layer (see Section 4.2). The linear correlation between the ice-water heat flux and

5   $\varepsilon$ supports the scaling (20) in the stratified boundary layer under ice: From the two bulk relationships Eq. (18) and Eq. (20), the former suggests $Q_{cw} \propto \varepsilon^{1/3}$ and the latter agrees with the observed linear dependence $Q_{cw} \propto \varepsilon$. Herewith, the result discards the widely used bulk relationship $Q_{cw} \propto u_*$ and suggests that $Q_{cw}$ scales as a friction velocity squared (Eq. 20) or, in terms of Eq. (19), $C_Q \propto u_*$. The dependence $Q_{cw} \propto u_*^2$ is well supported by our data, though the scaling is less apparent at very low $u_*$ due to the lack of values at low turbulence levels. Still, the data on $Q_{cw}(u_*)$ (Fig. 12b) clearly demonstrate the inappropriateness

10   of the "quasi-neutral" scaling (Eq. 18). Instead, the flux can be parameterized as

$$B_{iw} = 0.015\,u_*^2 N_S, \tag{25}$$

where $N_S$ is the quasi-constant buoyancy frequency in the boundary layer $0.4 < z < 10$ m.





## 5  Discussion

Our study presents the first detailed assessment of mixing conditions under the ice cover of Lake Baikal and their effect on the growth and melt of the ice cover. The seasonal ice cover is the Baikal's inherent feature, whose role in functioning of this unique ecosystem remains not fully understood. In this regard, the outcomes of the study underscore the importance of the

lake-wide circulation for the ice cover duration and ice thickness. The applicability of the results extends however beyond the specific Baikal conditions. Lake Baikal shares the major features of the seasonal ice cover on other lakes, as well as on inland and marginal seas allowing extension of the results on other ice-covered waters. Besides, the ice-water boundary layer in Lake Baikal owns a remarkable feature relevant to fundamental problems of environmental fluid mechanics: A strong boundary-layer flow on the background of permanent stable density stratification. In our study we successfully tested an alternative

approach to the traditional Monin-Obukhov similarity theory, based on the Dougherty-Ozmidov scaling. We also revealed several important facets of the turbulent energy budget in stratified boundary layers, as well as established a relationship between the shear turbulence under ice and the heat flux at the ice-water boundary.

The high values of water-ice heat fluxes in Lake Baikal and their apparent relationship to the intensity of under-ice circulation were previously noted by Aslamov et al. (2014, 2017). In the present study, the measured fluxes reached up to $40\,\mathrm{W\,m^{-2}}$ at their

peaks, which is an order of magnitude higher than values reported from small lakes Kirillin et al. (2018b) and comparable to the highest reported values of oceanic fluxes to the ice at the melting stage (Peterson et al., 2017). Concurrent registration of fluxes, current velocities and dissipation rates of the TKE reveals the direct link between turbulence production by the velocity shear and ice growth (ablation). The finding contradicts to the conventional assumption on the major role of convection produced by the solar radiation penetrating the ice in the under-ice mixing of freshwater lakes. While radiatively-driven convection

is a prominent feature of freshwater lakes (Farmer, 1975; Yang et al., 2017; Volkov et al., 2018), and effectively mixes the upper water column of Lake Baikal in winter (Granin et al., 2000; Jewson et al., 2009), its effect on the boundary mixing and heat transport to the ice base appears to be restricted by the stable stratification in the relatively thin interfacial layer with water temperature increase from the melting point at the IWI to that of the convectively mixed layer (Kirillin et al., 2018b). As a result, the energy of convection partially dissipates within the convective layer and is partially spent for entrainment of

deeper waters at the base of the mixed layer (Mironov et al., 2002). The rate of the energy dissipation produced by convection in small lakes (Volkov et al., 2018) is about $10^{-9}$-$10^{-8}\,\mathrm{W\,kg^{-1}}$, which is roughly an order of magnitude lower than that measured in this study. Consequently, the turbulence budget in the boundary layer differs significantly from that reported in previous lake studies. The TKE production averaged over the entire observations period prevails over dissipation at $z < L_N$, while in the deeper part of the boundary layer $\varepsilon$ slightly exceeds the production. Herewith, the boundary mixing continuously

pumps turbulent energy downwards, contributing to destroying of stratification in the main stratified layer SL $\delta < z < h_S$. The opposing trend is created by solar radiation, which increases the temperature of the convectively mixed layer $T_m$ and creates by this the upward exponential decrease of temperature from $T_m$ to the freezing point $T_f$. The observed quasi-linear stratification in the SL is apparently a result of the two opposing forces leading to a nearly steady-state conditions characterized by a constant $N$ within the layer $\delta < z < h_S$.



A particular advance in estimation of the turbulent energy budget under ice was achieved by direct estimation of the TKE dissipation rate using the velocity structure method. By this, we (i) avoided applying of the Taylor's "frozen turbulence" hypothesis, which remains questionable at relatively low current velocities under ice, and (ii) were able to trace the vertical distribution of $\varepsilon$ across the boundary layer (at least, a part of it, covering $\sim 2$ m). On one hand, this extended information

allows better quantification of the turbulent structure, on the other hand, it poses some fundamental questions on the major forces behind the under-ice mixing and heat transport to be discussed below.

Close to the ice base, vertical profiles of the TKE dissipation rates decayed as $\varepsilon \propto z^{-1}$, supporting the scaling of the turbulent mixing length with the distance from the solid boundary, similar to the neutral or nearly-neutral conditions. This fact gives a solid background to estimation of the shear velocity $u_*$ from the mean velocity profiles: The latter method is often uncertain,

given both depth of the logarithmic layer and the ice roughness are not known a priori. However, the thickness of the layer with $\varepsilon \propto z^{-1}$, varies depending on the current speed and mixing intensity. At strong currents, dissipation followed the scaling across the entire depth of the high-resolution velocity measurements of 1.4 m; at $\varepsilon << 10^{-7}$, it reduced to several tens of centimeters. The maximum mixing length scales remarkably well against the Dougherty-Ozmidov length scale: using $N_S$ as the characteristic value for stratification, $L_N = 1.2$ m at $\varepsilon = 10^{-7}$ W kg$^{-1}$ and $L_N = 0.4$ m at $\varepsilon = 10^{-8}$ W kg$^{-1}$. Qualitatively

the result can be treated as follows: the shear at the ice base dominates in the turbulence production at distances from the ice base less than $L_N$, while farther from the source of the shear the stratification limits the size of the turbulent eddies. This structure is also supported by comparison of the friction velocities computed from the two mixing lengths, Eq. (7) and Eq. (13): $u_*$ tends to be overestimated by the Ozmidov scaling at high mixing rates close to the ice base, and becomes lower than that produced by z-scaling at larger distances from the ice. At $z \approx L_N$ both estimations provide equal results (see Fig. 8). In the TKE budget,

the depth $z > L_N$ is also a turning point, where Eq. (14) shows a close balance between turbulence production and damping.

Generally, the simplified model of the TKE balance Eq. (14) was well supported by the data. The discrepancies included the prevalence of the TKE production over the damping terms closer to the ice cover and a slightly lower production than the sum of the lost terms at $z > L_N$. The imbalance can be tentatively attributed to downward advection of the TKE. Besides, the balance was estimated in assumption of the constant buoyancy frequency $N$, neglecting a possibly stronger loss of the TKE

closer to the ice, where $N$ increases. We also neglected possible transport of turbulent energy produced by convection due to solar heating. The latter can however be assumed small: within the stratified layer, the radiation levels under ice produced the destabilizing buoyancy flux of only $\mathcal{O}(10^{-10})$ W kg$^{-1}$, and the convectively mixed layer was located several meters beneath the deepest point covered by measurements.

The proposed scaling of the TKE budget differs from the conventional MOST approach by using the Dougherty-Ozmidov

length scale instead of the Obukhov length scale, i.e. by replacing the surface buoyancy flux with the mean stratification as a major scaling variable. This alternative approach is convenient for analysis of observational data: in contrast to the surface fluxes, $N$ is easily measurable in oceanic and lake studies. Noteworthy, both scaling approaches have been shown to be interchangeable (Grachev et al., 2015). In the particular case of ice-covered waters, the D-O approach is also more physically sound than MOST, since the surface buoyancy flux does not dominate the turbulent conditions under ice, it is rather a result

of the upward heat transport from deeper waters. One of the derivatives of the D-O boundary layer scaling is the relation-



ship $\varepsilon \propto u_* N$, which also implies that the length scale $u_* N^{-1}$ can be used instead of $L_N$ without any basic changes in the model assumptions. This scaling was previously considered e.g. by (Zilitinkevich and Mironov, 1996) and can be preferable for testing and refining the model parameters if observations of $u_*$ are available at high resolution.

The presence of an ice-adjacent interfacial sublayer $0 < z < \delta$ is another remarkable feature of the shear-dominated IBL. The

thickness of the layer ($\delta \approx 0.2$ m) was close to the smallest estimate based on the solar radiation of the layer created between the fixed temperature at the ice base and a convectively mixed homogeneous layer beneath. Our temperature measurements were too scarce to trace the evolution of its thickness at variable current velocities. Some insight into the genesis of the layer can be however obtained by assuming the largest value of the Dougherty-Ozmidov length scale $L_N \approx 0.85$ m as the maximal thickness of $\delta$. In this case, the mean buoyancy frequency in the interfacial sublayer $N_\delta \approx 0.02\,\mathrm{s}^{-1}$. That leads to the buoyancy

Reynolds number $Re_b \approx 16$ and the gradient Richardson number $Ri_g \approx 0.4$. Both values are close to the critical values of $\mathcal{O}(10^1)$ and 0.25, respectively. The layer $0 < z < \delta$ may be therefore assumed to stay in near-critical turbulence-free state. In the rest of the boundary layer, both $Ri_g$ and $Re_b$ are far beyond the critical values, indicating developed turbulence.

The following structure can be tentatively drawn from the above analysis (Fig. 13): the background vertical temperature (density) distribution is formed by absorption of solar radiation and the upward heat flux to the ice, producing the profile

typically observed in ice-covered lakes without strong currents: a homogeneous convectively mixed layer with a relatively thin stratified layer on top. Mixing by the velocity shear reduces the density gradient. In the top layer, shear mixing is balanced by solar heating from above, so that the density gradient tends to the critical value between turbulent and non-turbulent states. In the upper part of the homogeneous layer, a weaker nearly linear density gradient forms. In the larger part of this layer, at $z > L_N$, the turbulence production is balanced by the stratification, in accordance with "z-less" linear scaling, Eq. (13).

Beneath the depth $L_N$, the scaling suggests $\partial U \partial z^{-1} = const$ that is also supported by the measured mean velocities (Fig. 8). The stratification is nearly linear suggesting also that $\varepsilon$ is nearly constant across it. The thickness of the layer depends apparently on the scales of the under-ice flow. We did not consider the rotational effects on the boundary layer characteristics. A slight Ekman-like rotation of the mean current was observed under ice (Fig. 3), and the Ekman scale $u_* f^{-1}$ based on the mean shear velocities is $\mathcal{O}(10^1)$ m. Herewith, the Coriolis force may have an effect on the thickness of the boundary layer. At weak

stratification, effects of both $N$ and $f$ on the boundary layer dynamics may appear comparable, so that the model will perform better if the length scale $L_N$ (or its equivalent $u_* N^{-1}$) is replaced by the combined length scale $u_*(fN)^{-1/2}$. The latter scaling was proposed by (Pollard et al., 1972). Zilitinkevich and Mironov (1996) suggested that the $fN$-scaling has rather limited area of application. However low flow velocities and weak stratification are typical for conditions under ice, and the Pollard et al. (1972)-scaling may find its application in ice-covered waters. More data are required here, in particular, on the fine density and

flow structure over the entire Ekman layer.

The main motivation of the study was seeking for the relationship between the shear mixing and stratification on one hand and the heat release from water to the ice cover on the other hand. In this regard, the scaling of the heat (buoyancy) flux with the shear velocity squared and the buoyancy frequency under ice (Eq. 4.5) is our major finding with implications to a wide range of the ice-related problems. The simple "bulk" approximation Eq. (19) is widely used in models of ice-water interaction,

but its validity was never thoroughly tested before. The simple relationship $Q_{cw} \propto u_* \Delta T$, equivalent to Eq. (19), failed to



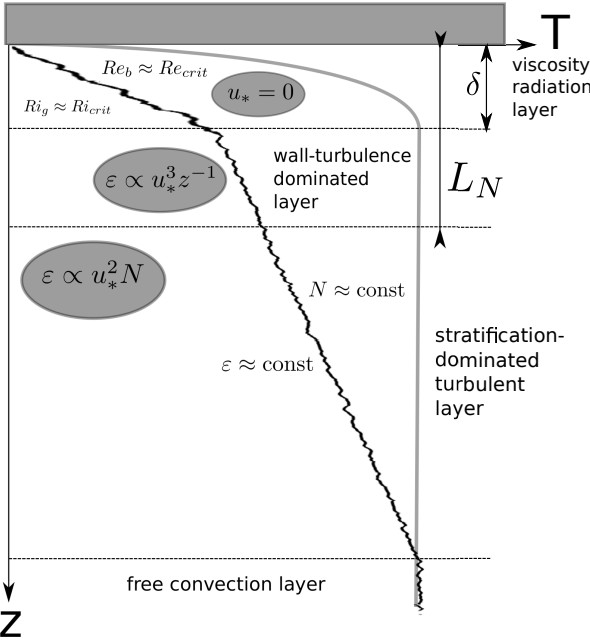

**Figure 13.** Formation of the vertical temperature profile in a shear-driven ice boundary layer. Gray line is a no-shear original profile, black line is the result of shear-driven turbulence on the background of stable stratification. See text for notations.

describe the heat flux dynamics in the ice boundary layer, replaced by the scaling $Q_{cw} \propto (g\alpha_T)^{-1}u_*^2N$. The result suggests that the effect of the currents on the decrease of the arctic sea ice may appear much stronger than assumed by the present model projections, based on the bulk estimation $Q_{iw} \propto u_*$. Further decline of the arctic ice cover may result in both increase of the under-ice current speeds due to changed global circulation and increase of the stratification due to warming of the under-ice

5    waters. Both factors, according to our scaling will accelerate the vertical heat transport from water to the ice, resulting in a positive feedback on the ice melt.

In contrast to the surface heat flux, the bulk formulation Eq. (6) worked well for the momentum flux thanks to the nearly constant shear conditions close to the ice base. The result is of practical use in simple models of ice covered seas and lakes, where $u_*$ can be directly approximated from the mean current speeds at a certain distance from the IWI (Fig. 10). The speeds

10   should be however known at distances from the ice $z < L_N$, otherwise, the stratification effects on turbulence production make the bulk formula not representative.

## 6   Conclusions

We investigated the fine vertical structure of turbulence characteristics in the boundary layer of Lake Baikal and proposed a model of stratified turbulent ice boundary layer based on the Dougherty-Ozmidov length scale of turbulence. The ultimate





result consists in scaling of the water-ice heat flux against the shear velocity squared. The result suggests large errors in the heat flux estimations, when the traditional "bulk" approach is applied to stratified conditions with strong shear. It also implies that under-ice currents may have much stronger effect on the ice melt than estimated by traditional models.

*Data availability.* Data are available from the authors by request

5 *Author contributions.* GK, IA and NG conceived the study; IA, RZ, and GK designed and performed the field experiments; GK developed the model; GK, IA, VK, and NG performed field data analysis; GK wrote the manuscript with contributions from all co-authors

*Competing interests.* The authors declare no competing interests

*Acknowledgements.* The study is a part of the research project "IceBound" supported by the German Research Foundation (DFG Project KI-853/11-1, KI-853/11-2). Additional support by the DFG Project KI-853/13-1 is gratefully appreciated by GK.





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
