# Peer review of "Turbulence in the stratified boundary layer under ice: observations from Lake Baikal and a new similarity model"

_Hydrology and Earth System Sciences, 2019_

## Referee Comment (RC1) · Anonymous Referee #1 · 5 Jan 2020

1. Scientific Significance: Good Kirillin et al. 2019 presents an interesting approach to problems encountered in classic turbulence theory when TKE production is substantially affected by interfacial buoyancy fluxes. The authors' argue that the TKE balance, and associated fluxes, are inappropriately characterized by traditional bulk parameterizations for dynamic (sheared) and static (convective) instabilities. This is a challenging scientific problem in need of a solution and ranks among the largest challenges for numerical modelers. The authors make a compelling argument that the DO scaling approach (instead of the classic MO scaling) is practical for boundary layers generated by flows under freshwater lake ice.

[Figure]

2. Scientific Quality: Excellent Pg. 2 Line 8: Reference in manuscript about sea ice loss attributed "primarily" to basal ice melt (ocean-to-ice) as opposed to surface ice melt (air-to-ice) is still under debate. Ice mass balance (IMB) observations shows that the amount of surface (atmospheric) and basal (oceanic) melt varies with each year (some years the top melts about the same as the bottom). I recommend rewording this sentence to state that a significant component of sea ice volume loss occurs from the sea ice bottom... (or something like "due to ocean-to-ice heat fluxes").

Pg. 2 Lines 16-28: Great discussion in this paragraph about the uniqueness of under-ice temperature stratification under lake ice (different than my background in sea ice and salinity driven density gradients); however, I'm somewhat confused on the persistence of the interfacial layer (IL) during SML free convection. Results from Frey et al. (2011) and Light et al. (2008) show that significant solar radiation is deposited immediately beneath the sea ice; therefore, how is the IL maintained if the strongest heating (solar) is in the layer closest to the ice base. I assume this is due to either high sensible heat losses caused by the negative ice temperature gradients (thermal conductivity), or latent heat losses to the lake ice base (or combination of both in March); either way, the negative heat budget despite solar heating near the ice-water interface should briefly be addressed here. I also see that you discussed the near ice heat budget on pages 3 and 4; perhaps the best solution in the intro is to capture the dominating heat loss term during the period of your study (latent heat or sensible heat).

Pg. 2 Lines 21-23: In Arctic Ocean air-ice-water interactions, entrainment of subsurface heat (usually the near-surface temperature maximum (NSTM)) is hard to achieve with static instabilities (e.g. brine rejection), this is usually reserved for stronger dynamic (sheared) instabilities. If this statement (lake heat entrainment with static instabilities) has been demonstrated by previous work, please reference.

Pg. 3 Section 2: Regarding the geostrophic currents in Lake Baikal, request there be some background provided in this section as to the source of this current (pressure gradient force created by???? and spatial scale drives a low Rossby Number environment, etc.).

Pg. 8 Lines 17-18: Why were there more current meters deployed at S1 and not at S2?

Pg. 10 Figure 2: Perhaps I missed this in the results discussion, but why did the penetrated solar radiation drop off substantially after Feb. 22 when topside solar radiation increased (Figs 2c and 2d). Was there a snow event(s)? The only reference I can find to Fig. 2d is on page 13 and only accounts for the mean daily under-ice short-wave radiation (Io = 9.7 W mˆ-2) and a range of through-ice radiation (8-18%). These results do not match with the results in Figs. 2c and 2d where the transmissivity (solar(under-ice)/solar(top-ice)) between March 6th and March 16th appears to be well below 1% with under-ice radiation values of <2 W mˆ-2. The extremely low under-ice radiation values heavily skew the 9.7 W mˆ-2 average over the study period and likely affects the intensity of short-wave induced convective overturning in the SML. There appears to be two "modes" to this dataset: 1) light snow cover prior to 22 February with active SMLs and strong ILs; and 2) moderate-heavy snow cover after 22 Feb with inactive SMLs and weak ILs. Request clarification on how this transition in the steady state condition was handled and why it is appropriate to conduct DO scaling model validations across these varying conditions.

Pg. 17 Lines 2-4: Once again, it appears that the event (likely snow) heavily impacted these results during the 24 Feb – 07 Mar period. If the 22 Feb event is snow, I anticipate it would affect several areas of the heat budget and near interface buoyancy to include lowering ice-to-air sensible heat fluxes and destabilizing the IL (less downwelled solar radiation) allowing turbulent (shear) eddies access to the ice base. If this were indeed the case, it should probably be integrated into the discussion, if not, recommend addressing the cause of the significant change in heat balance conditions in Fig. 11 after 22 Feb (similar to the previous comment for page 10).

Pg. 19 Lines 14-16: Not entirely accurate, oceanic fluxes during the 2014 MIZ experiment in the Beaufort Sea were >100 W mˆ-2 with Autonomous Ocean Flux Buoys

(Gallaher et al., 2016) and nearly 200 W m^-2 in the Greenland Sea during the 1983/84 MIZ experiment (McPhee et al., 1987).

Pg. 21 Line 6: I did not see an isothermal/isopycnal (homogeneous) layer mixed layer in the data (Fig. 3); perhaps, near-homogeneous is more appropriate.

Pg. 22 Line 16: Interesting idea to scale this DO scaling approach to sea ice modeling; however, near-freezing freshwater and seawater ice-water boundary layers have notable differences. Things that come immediately to mind are: 1) the rotational Ekman layer plays an important role (which was not tested in your study) in the deeper dynamically developed ocean boundary layers (20-35 cm/sec free drift ice speeds); 2) temperature becomes the equivalent of a passive tracer (no buoyancy contribution) in seawater above ∼25 ppt; and, 3) bulk parameterizations using MO scaling have worked pretty well when validated against eddy correlation and thermal dissipation observations. I will admit, that during calm wind conditions in the presence of significant meltwater (melt pond drainage), this parameterization does not perform well and is similar to your study minus the temperature stratification from solar heating. For this paragraph, I would recommend rewriting to target the potential benefit of this approach during weak atmospheric forcing over sea ice during the melt season.

Pgs. 22-23 Section 6: Conclusion seems a little abbreviated, recommend recapping a few more of your findings.

3. Presentation Quality: Excellent Manuscript figures are good quality and well labeled. I only have a few recommendations regarding data visualization: Pg. 8 Figure 1: Recommend including the type of satellite imagery used (visible, LandSat, IR, SAR...)

Pg. 8 Figure 1: This is not strictly required, but I recommend including another figure (or figure inset) that shows the general geographic location of Lake Baikal. Because I'm not familiar with the Lake, I had to look online for some spatial context and whether it was a terrestrial (continental interior) or maritime (near coastal plain) lake.

4. Technical Comments: Pg 1 Line 10: Not sure "the latter" is required since only one topic is being referenced Pg 2 Lines 28-29: Recommend runoff or accelerated sea ice melt... Pg 3 Line 2: Recommend high water to ice heat fluxes in Lake Baikal... Pg. 3 Lines 3-5: I don't completely understand this sentence, recommend rewriting for clarity. Pg. 3 Line 15: analyze the effect of turbulent mixing... Pg. 3 Line 16: study is to establish the scaling... Pg. 3 Line 17: circulation with the seasonal ice cover dynamics and suitable parameterization for ice-water heat exchange... Pg. 3 Line 24: and markedly increase... Pg. 3 Lines 24-25: Similarly, the increase... flow can destroy the conduction layer... Pg. 3 Line 26: In the majority of freshwater lakes, the aforementioned... Pg. 3 Line 29: impurities than sea ice or river ice... Pg. 4 Line 18: does not typically... Pg. 4 Line 21: I don't understand how TKE can be supplied by the "decay of the convective motions." Shouldn't this be "by the displacement of the underlying water by density-driven static instabilities." Recommend a little further clarification in this sentence. Pg. 4 Line 31: roughness length (instead of parameter)? Pg. 5 Line 4: bulk transfer or drag coefficient... Pg. 5 Line 6: tends to be the local balance... Pg. 5 Lines 10-11: This sentence is hard to follow, perhaps something like the "the second factor influencing near boundary stratification is the destabilizing buoyancy flux due to... water column of thickness hs and is derived from..." If this isn't what you mean, recommend rewording for clarification. Pg. 6 Line 17: production of TKE is limited by two major loss processes... Pg. 8 Line 10: Three short-wave radiation sensors were deployed vertically to measure the decay of solar radiative fluxes across the air-ice-water system. Pg. 9 Line 26: while basal ice at Station S2 continued to grow at a slow rate of ~~0.3 cm dayˆ-1... Pg. 9 Line 26: I believe the Figure reference should be 2b not 2c. Pg. 15 Line 32: made by the single-point Pg. 17 Line 8: Long dashed line to start this line, I believe this should be deleted. Pg. 23 Line 3: currents may have a much stronger effect on lake ice melt than estimated by...

Please also note the supplement to this comment:
https://www.hydrol-earth-syst-sci-discuss.net/hess-2019-608/hess-2019-608-RC1-

supplement.pdf

---

## Short Comment (SC1) · 7 Jan 2020

The present manuscript presents a nice field work and parameterizing scheme for the heat flux (Fw) from water to ice bottom, which needs more efforts to deal with comparing to other heat fluxes involved in seasonal ice cover thermodynamics. This work approved that, in large lakes, under-ice currents (e.g. lake circulation, down-slope current, and seiche-induced turbulence in authors's previous papers) take significant impacts on under-ice mixing and thus increase the water-to-ice heat flux largely, much larger than reported values in Arctic/temperate lakes. But I have to say, such large Fw was also reported just recently in a small shallow Qinghai-Tibet Plateau thermokarst

pond in Huang et al (2019, Thermal structure and water-ice heat transfer in a shallow ice-covered thermokarst lake in central Qinghai-Tibet Plateau, Journal of Hydrology, 2019, 578, 124122. https://doi.org/10.1016/j.jhydrol.2019.124122), where the Fw was estimated both from heat residual method of bottom ice layer and from heat balancing within under-ice water layer. Intensive solar radiation and under-ice vertical and horizontal currents may be the reason. And modeling experiments were also performed in term of different Fw schemes and demonstrated that the current simple schemes for Fw can not give agreeable results on ice thickness and temperature (Huang et al., Modeling experiments on seasonal lake ice mass and energy balance in Qinghai-Tibet Plateau: A case study, Hydrology and Earth System Sciences, 2019, 23(4): 2173-2186, doi:10.5194/hess-23-2173-2019). so, here, is it possible to esimate the Fw based on heat balancing in water layer?

---

## Referee Comment (RC2) · Anonymous Referee #2 · 1 Feb 2020

General Comments

The authors present compelling evidence – both theoretical and observational – for a new boundary layer model of flow beneath the ice cover of a freshwater lake. The approach taken is relevant for many large lakes with significant boundary currents, and in some cases may be important for sea ice as well: as such this will be of interest for readers of HESS. Nevertheless the manuscript would benefit from some clarification as outlined below.

Scaling Arguments

The theoretical development (§2) is relatively straightforward but could be more clear.

It seems a bit unfortunate that both Qcw and Qiw are used for the water-ice heat flux. I expect Qcw is used to emphasize that the flux is purely conductive (eqn. 2). Later it is shown that the flux is more than conductive, as exemplified by the turbulent diffusivity K in the equation between eqn. 19 and eqn. 20. It would be more clear to use Qiw throughout, and argue that the new scaling shows that the diffusivity in eqn. 2 should be turbulent and not molecular. Thus the water-ice heat flux will be larger, and the ice at station S1 is thinner. Some additional minor points for this section: It would be nice to see a reference for eqn. 8. If it is Mironov et al. then state this before you write the equation. Also, $\nu$ is the kinematic viscosity – not the viscosity though it is clear what you meant (pg. 7, line 7).

Results

Fig. 3 has no time/date indicated. Are these data averaged over some period or a single snapshot? Also, there are important differences between Fig. 3a (station S1-strong flows) and Fig. 3b (S2- weaker flows) that are not discussed. It is pointed out the S1 shows 2 stratified layers above the mixed layer below 10 m depth – but S2 clearly shows 3 layers, evidently related to some process near the ice-water interface. Both are different from the canonical no-flow situation, exemplified in Fig. 13 (gray curve) which has 1 stratified layer. So as flows are increased from zero to strong the number of stratified layers seems to increase from 1 to 3 and then fall to 2. Can you explain this? Also, one assumes the total heat content in Fig. 3b (thicker ice) is larger than in Fig. 3a (thinner ice) – it might be of interest to compute this, comparing to the latent heat of the ice cover difference.

Other minor points:

Pg. 10, L.2-6: This paragraph appears to describe S1, though some points are relevant to both stations. You should clarify. Pg. 15, L. 18: Fig. 9b should be Fig. 9a. Likewise, on L.23 Fig. 9a should be Fig. 9b. Pg. 21, L.33: Eq. 4.5 should be Eq. 20.

[Figure]

608, 2019.

---

## Author Comment (AC1) · 28 Feb 2020

**We thank the Reviewer for the careful reading of our study and the thoughtful comments. We particularly welcome the comments from the oceanographic point of view, which help to clarify the similarities and differences between freshwater and oceanic ice-covered systems. The discussion will surely help our study to reach the oceanographic audience, and enable implementation of our results on wider scales. Below is the point-to-point reply to the comments.**

Pg. 2 Line 8: Reference in manuscript about sea ice loss attributed "primarily" to basal ice melt (ocean-to-ice) as opposed to surface ice melt (air-to-ice) is still under debate.

Ice mass balance (IMB) observations shows that the amount of surface (atmospheric) and basal (oceanic) melt varies with each year (some years the top melts about the same as the bottom). I recommend rewording this sentence to state that a significant component of sea ice volume loss occurs from the sea ice bottom... (or something like "due to ocean-to-ice heat fluxes")

**When pointing on the primary role of the basal melt, we referred to:**

*Carmack, et al.: Toward quantifying the increasing role of oceanic heat in sea ice loss in the new Arctic, Bulletin of the American Meteorological Society, 96, 2079-2105, 2015.*

**We appreciate that uncertainty may still exist in the estimates of the relative role of surface and basal melting, and rephrase the sentence in a less categorical way, as suggested.**

Pg. 2 Lines 16-28:

I'm somewhat confused on the persistence of the interfacial layer (IL) during SML free convection. How is the IL maintained if the strongest heating (solar) is in the layer closest to the ice base. I assume this is due to either high sensible heat losses caused by the negative ice temperature gradients (thermal conductivity), or latent heat losses to the lake ice base (or combination of both in March); either way, the negative heat budget despite solar heating near the ice-water interface should briefly be addressed here...perhaps the best solution in the intro is to capture the dominating heat loss term during the period of your study (latent heat or sensible heat).

**The raised issue is indeed quite important for understanding the under-ice boundary layer dynamics and is characteristic of fresh (and brackish) waters. In short: The stable density stratification prevents convective mixing despite the negative buoyancy production by the decrease of the solar radiation with depth.**

**Thanks to the freshwater density anomaly, density increases with temperature**

**in fresh and brackish waters at temperatures below the maximum density point ($\approx 3.98\ ^o$C for freshwater). At low salinities, water temperatures remain always higher than that of ice; hence the stably stratified interfacial layer (IL) with downward temperature increase is inevitable under the ice base. As a result, the heat budget in the IL is governed by the balance between the radiation absorption and—as Reviewer rightly suggested—upward heat conduction to the ice base. The balance was considered by Barnes and Hobbie (1960, referred in the paper), who proposed an elegant analytical solution of the heat transport equation for the thickness of the IL, temperature profile within it, and the water-ice heat flux in purely conductive conditions (see also Mironov et al. 2002, referred in the paper, for an extended discussion). Noteworthy, the IL is not purely conductive even in small lakes, characterized by intermittent wave-generated turbulence (Kirillin et al. 2018, referred in the paper), making the estimation of the ice-water heat exchange in the absence of mean flow particularly challenging. The deformation of IL in the presence of a strong mean flow is considered in Discussion section of the paper. We add a reference to the conduction-radiation model of Barnes and Hobbie to the introduction.**

Pg. 2 Lines 21-23: In Arctic Ocean air-ice-water interactions, entrainment of subsurface heat (usually the near-surface temperature maximum (NSTM)) is hard to achieve with static instabilities (e.g. brine rejection), this is usually reserved for stronger dynamic (sheared)instabilities. If this statement (lake heat entrainment with static instabilities) has been demonstrated by previous work, please reference.

**Convection due to solar heating in freshwaters is more energetic than mixing by brine rejection and represents a classical natural example of penetrative (entraining) convection in the absence of the mean flow shear with analogies in atmospheric, oceanic and astrophysical flows. The first detailed analysis was performed by Farmer (1975, referred in the paper), a review is given by Kirillin et al. (2012, referred in the paper).**

Pg. 3 Section 2: Regarding the geostrophic currents in Lake Baikal, request there be some background provided in this section as to the source of this current (pressure gradient force created by???? and spatial scale drives a low Rossby Number environment, etc.)

**Being meticulous, we did not mention a geostrophic character of the observed currents in Section 2. Still, since Lake Baikal in winter is completely isolated from the direct contact with the atmosphere by the ice cover, the pressure (density) gradient is the major driver of under-ice flows. The characteristic velocity and spatial scales of the observed under-ice currents are ($10^{-2}$-$10^{-1}$ m s$^{-1}$) and ($10^4$-$10^5$ m), respectively (Figs. 1 and 5 of the paper). Hence, away from boundaries it is balanced mainly by the Coriolos force. The under-ice current of the same scales is persistent in this part of Lake Baikal (Aslamov et al. 2014, 2017, cited in the paper). Therefore, the geostrophy was mentioned in the abstract of the paper. The origin of the pressure gradient forcing is not exactly known: large-scale density fields under the Baikal ice were not measured. The usual suspects are the the wind-topography interactions, which create a regular large-scale snow cover pattern on the ice surface. As a result, the horizontal temperature gradients in under-ice waters are created by the inhomogeneous heating by solar radiation, as documented by Aslamov et al. (2017, cited in the paper). We add this information to the revised version and remove the only use of the word "geostrophic" from the abstract.**

Pg. 8 Lines 17-18: Why were there more current meters deployed at S1 and not at S2?

**The available amount of loggers did not allow to obtain a detailed vertical resolution at both stations. Therefore, bulk of the loggers were deployed at the primary site, with only one logger deployed at the reference site to background the AQUADOPP measurements.**

Pg. 10 Figure 2: Perhaps I missed this in the results discussion, but why did the penetrated solar radiation drop off substantially after Feb. 22 when topside solar radiation increased (Figs 2c and 2d). Was there a snow event(s)? The only reference I can find to Fig. 2d is on page 13 and only accounts for the mean daily under-ice short-wave radiation ($I_0$ = 9.7 W m$^{-2}$) and a range of through-ice radiation (8-18%). These results do not match with the results in Figs. 2c and 2d where the transmissivity (solar(under-ice)/solar(top-ice)) between March 6th and March 16th appears to be well below 1% with under-ice radiation values of <2 W m$^{-2}$. The extremely low under-ice radiation values heavily skew the 9.7 W m$^{-2}$ average over the study period and likely affects the intensity of short-wave induced convective overturning in the SML. There appears to be two "modes" to this dataset: 1) light snow cover prior to 22 February with active SMLs and strong ILs; and 2) moderate-heavy snow cover after 22 Feb with inactive SMLs and weak ILs. Request clarification on how this transition in the steady state condition was handled and why it is appropriate to conduct DO scaling model validations across these varying conditions.

**It is a valuable comment for specifying the background conditions behind the under-ice boundary layer formation. Indeed, the drop of the under-ice solar radiation was caused by a (relatively light, ≈0.5 cm) snowfall, which prevented the light penetration through the otherwise transparent congelation ice. As correctly mentioned by the reviewer, variations in the under-ice radiation could have affected the temperature distribution under ice by slowing down or even canceling the warming in the convectively mixed layer at depths below 10 m. However, the estimates of the convective velocities $w_*$ (Section 4.3) demonstrated that the radiation was of minor importance for the mixing conditions in the boundary layer compared with the shear instabilities $u_*$. Also, the background stratification did not change much during the period of observations because of the large thickness of the nearly homogeneous "free convection layer" and the strong effect of the shear mixing on the stratification above it (see Fig. 13 and the discussion around it). Therefore, shear and stratification in the boundary layer remain to be the major factors determining water-ice heat transport, making the D-O scal-**

ing to the ideal choice for parameterization of boundary fluxes. When modeling
fluxes at longer time scales, changes in the background stratification caused by
radiation variability are dirrectly accounted for in the scaling by variations in the
buoyancy frequency $N$.

**We mention this fact in the revised paper and adjust the lower value of the sub-
surface radiation.**

Pg. 17 Lines 2-4: Once again, it appears that the event (likely snow) heavily impacted
these results during the 24 Feb – 07 Mar period. If the 22 Feb event is snow, I antic-
ipate it would affect several areas of the heat budget and near interface buoyancy to
include lowering iceto-air sensible heat fluxes and destabilizing the IL (less downwelled
solar radiation) allowing turbulent (shear) eddies access to the ice base. If this were in-
deed the case, it should probably be integrated into the discussion, if not, recommend
addressing the cause of the significant change in heat balance conditions in Fig. 11
after 22 Feb (similar to the previous comment for page 10).

**Similar to the previous comment, the remark correctly refers to the synoptic vari-
ability, which was only briefly mentioned in the original manuscript: apart from
reducing the under-ice solar radiation, the snow cover reduced the heat release
at the ice surface (the ice surface heat budget was not a subject of our study,
though). As a result, the conductive heat flux at the ice base $Q_{ci}$ reduced, while,
as we mentioned in the paper, remained positive, with values up to 40 W m$^{-2}$.
The major point here is: the turbulence due to the mean flow produced water-ice
heat fluxes was sufficient to initiate ice melt at its base (cf. the negative peak in
the fluxes in Fig. 11a and the velocity peak in Fig. 5a on 06-07.03.2017, Station 1).**

**We amend the sentence to clarify this issue.**

Pg. 19 Lines 14-16: Not entirely accurate, oceanic fluxes during the 2014 MIZ ex-
periment in the Beaufort Sea were >100 W m$^{-2}$ with Autonomous Ocean Flux Buoys
(Gallaher et al., 2016) and nearly 200 W m$^{-2}$ in the Greenland Sea during the 1983/84

MIZ experiment (McPhee et al., 1987)

**The comment echoes the remark of Wenfeng Huang (see our response in the HESSD discussion). Our sentence is however formulated in a quite accurate way: the fluxes we reported are indeed significantly higher than those in small ice-covered lakes and are comparable to those reported in the ice-covered seas. Shallow alpine thermokarst ponds and drifting ice during strong storms in the ocean are the extreme examples worth mentioning, so we add references to Huang et al. (2019) and Gallaher et al. (2016).**

Pg. 21 Line 6: I did not see an isothermal/isopycnal (homogeneous) layer mixed layer in the data (Fig. 3); perhaps, near-homogeneous is more appropriate.

**Agreed. "Nearly homogeneous" is added.**

Pg. 22 Line 16: Interesting idea to scale this DO scaling approach to sea ice modeling; however, near-freezing freshwater and seawater ice-water boundary layers have notable differences. Things that come immediately to mind are: 1) the rotational Ekman layer plays an important role (which was not tested in your study) in the deeper dynamically developed ocean boundary layers (20-35 cm/sec free drift ice speeds); 2) temperature becomes the equivalent of a passive tracer (no buoyancy contribution) in seawater above 25 ppt; and, 3) bulk parameterizations using MO scaling have worked pretty well when validated against eddy correlation and thermal dissipation observations. I will admit, that during calm wind conditions in the presence of significant meltwater (melt pond drainage), this parameterization does not perform well and is similar to your study minus the temperature stratification from solar heating. For this paragraph, I would recommend rewriting to target the potential benefit of this approach during weak atmospheric forcing over sea ice during the melt season.

**A very appreciated while predictable comment. We agree that the M-O scaling would work in ice-covered seas at appreciable shear mixing, especially during ice growth. The D-O scaling would generally work in this case too, since both**

scalings can be derived from each other (see Grachev et al. 2015, cited in the paper). During the melting phase, in turn, the traditional M-O scaling can produce significant errors in the rates of basal ice melt, especially at strong under-ice stratification. Besides, the D-O scaling has several theoretical and practical advantages: Stratification in the boundary layer, not the buoyancy flux at the ice base, is the mechanism directly damping the turbulence under ice that makes $N$ to a "natural choice" for scaling. Also, D-O scaling is easily applicable in practice: stratification is directly obtained from observations/models, which is often difficult with the boundary buoyancy flux. We use this opportunity to argue briefly, why the D-O scaling can be advantageous for modeling of the ice-ocean boundary layer:

(1) We have outlined in our Discussion how the Ekman forcing can be incorporated in the D-O scaling (Page 21, Lines 22-30). It does not seem to be a crucial issue however: The conventional M-O scaling does not include the Coriolis force either. See Zilitinkevich and Mironov (1996, referred in the paper) for a discussion on the scales, where the Coriolis forcing is important for the boundary layer scaling.

(2) Even if temperature has no contribution to the buoyancy in saline waters, the sea-ice boundary layer during the melt phase tends to be strongly stratified due to freshening (e.g., by basal ice melting, freshwater inflows, or melt pond drainage, as mentioned by the Reviewer), so that $N$ becomes to the major factor dumping the shear-produced turbulence at the ice-water interface. The M-O scaling is apparently less relevant in this case.

(3) The neutral (M-O) scaling worked fine for the *momentum* fluxes in Lake Baikal too (in our Discussion we propose an explanation why). The impetus for the development of the new scaling behind the boundary-layer modeling was provided by the apparent inconsistency of the M-O scaling for the *scalar* (heat) fluxes, crucial for correct estimation of the ice melting rates. We strongly believe the

**scaling is more physically sound in ice-covered seas and is, at least, worth trying in marine models.**

Pgs. 22-23 Section 6: Conclusion seems a little abbreviated, recommend recapping a few more of your findings.

**We extend the conclusions taking into account the HESSD discussion.**

Minor comments

**We incorporated all suggestions in the revised version**

––––––––––––––––––––

---

## Author Comment (AC2) · 28 Feb 2020

We sincerely thank Dr. Huang for the valuable contribution to the discussion on our study. Indeed, the recently appeared study of Huang et al. (2019) on heat budget of a Tibetan ice-covered pond represents a great evidence of the large importance of the water-ice heat flux for the ice cover growth and melt: The absolute values of the flux in a thermokarst alpine lake are comparable or even larger than those found in Lake Baikal that makes the heat balance of two systems similar. Both in our study and in the study of Huang et al. (2019), the water-ice heat flux was estimated from the heat balance at the ice-water interface, using similar measurement setup with frozen thermistor chains. In that way, the results are inter-comparable, and we add in Discussion a reference to the study of Huang et al. The possible way of parameterization of the ice-water heat flux in thermokarst Tibetan lakes remains to be an intriguing question. The D-O scaling proposed in our study can only be used used if information is available on the TKE production or dissipation rates, with subsequent parameterizations of the turbulence intensity based on the large-scale forcing. The latter is apparently different in thermokarst lakes than in Lake Baikal or other large lakes and seas. In deep Lake Baikal, in turn, the water column remains relatively cold throughout the ice season, but the strong upward heat flux is conditioned by the strong turbulent mixing due to large-scale under-ice currents. The upward heat release in the rather shallow ($< 2$ m mean depth) thermokarst lake is produced mainly by the heating of the water column due to the solar radiation, which is very strong over the Tibetan Plateau. As a result, the water temperatures under ice achieve values of up to 9 $^o$C, creating a strong gradient at the ice-water interface. The apparent source of turbulence is in this case free convection due to the negative buoyancy flux. In the typical conditions during the ice cover melt in Tibetan lakes, the mean water column temperature is above the maximum density values of $\approx 3.98^o$C. In this case, the convection is very non-stationary (see e.g. Kirillin and Terzhevik 2011), so that its parameterization is non-trivial. As rightly mentioned by Huang et al. in their discussion, direct measurements of turbulence under ice could provide the necessary quantitative information on the turbulence mixing in these quite specific conditions.

———————————————————

---

## Author Comment (AC3) · 28 Feb 2020

We thank sincerely the Reviewer for the careful review and valuable suggestion on our study, which helped to improve the manuscript. Below we reply to the Reviewer's comments, providing only titles from original review:

**Scaling arguments**

- Both $Q_{cw}$ and $Q_{iw}$ were used interchangeably in the original version, which can be of course misleading to the reader - thank you for mentioning it. We have consistently changed all $Q_{cw}$'s with $Q_{iw}$.

[Figure]

- Mironov et al. (2002) is cited right after Eq. (8)

- We added "kinematic" to viscosity for clarity.

**Results**

- Interesting questions are raised here about the relationship between the flow strength and the temperature profile in the upper 10 m of the water column. Qualitatively, the profiles at both sites are closer to straight lines than to an exponential "zero-shear" profile from Fig. 13, with the profile at lower current speeds (S1, Fig.3a) is slightly "convex", i.e. closer to the theoretical curve from Fig. 13 and stronger currents correspond to a slightly "concave" shape of the temperature profile (S1, Fig.3a). In our discussion (Page 21, Fig. 13), we propose a nearly steady-state balance between stratification and shear as an explanation of the quasi-linear temperature profile. The deviation from the straight line, and as a result, the variations in the heat content of the upper water column could be caused by horizontal advection of heat by currents, or by restratification during periods of low shear. We do not possess enough well-resolved data on temperature to investigate these effects. We clarified this point in the text and also added time/date to Fig. 3.

**Other minor points**

Thankfully accepted.